# A Geometric Perspective on Variational Autoencoders

**Clément Chadebec**
Université Paris Cité, INRIA, Inserm, SU
Centre de Recherche des Cordeliers
clement.chadebec@inria.fr

**Stéphanie Allassonnière**
Université Paris Cité, INRIA, Inserm, SU
Centre de Recherche des Cordeliers
stephanie.allassonniere@inria.fr

## Abstract

This paper introduces a new interpretation of the Variational Autoencoder framework by taking a fully geometric point of view. We argue that vanilla VAE models unveil naturally a Riemannian structure in their latent space and that taking into consideration those geometrical aspects can lead to better interpolations and an improved generation procedure. This new proposed sampling method consists in sampling from the uniform distribution deriving intrinsically from the learned Riemannian latent space and we show that using this scheme can make a vanilla VAE competitive and even better than more advanced versions on several benchmark datasets. Since generative models are known to be sensitive to the number of training samples we also stress the method's robustness in the low data regime.

## 1 Introduction

Variational Autoencoders (VAE) [25, 43] are powerful generative models that map complex input data in a much lower dimensional space referred to as the latent space while driving the latent variables to follow a given prior distribution. Their simplicity to use in practice has made them very attractive models to perform various tasks such as high-fidelity image generation [41], speech modeling [5], clustering [52] or data augmentation [8].

Nonetheless, when taken in their simplest version, it was noted that these models produce blurry samples on image generation tasks most of the time. This undesired behavior may be due to several limitations of the VAE framework. First, the training of a VAE aims at maximizing the Evidence Lower BOund (ELBO) which is only a lower bound on the true likelihood and so does not ensure that we are always actually improving the true objective [6, 1, 21, 13, 53]. Second, the prior distribution may be too simplistic [14] leading to poor data generation and there exists no guarantee that the actual distribution of the latent code will match a given prior distribution inducing distribution mismatch [12]. Hence, trying to tackle those limitations through richer posterior distributions [46, 42] or better priors [50] represents a major part of the proposed improvements over the past few years. However, the tractability of the ELBO constrains the choice in distributions and so finding a trade-off between model expressiveness and tractability remains crucial. In this paper, we take a rather different approach and focus on the geometrical aspects a vanilla VAE is able to capture in its latent space. In particular, we propose the following contributions:

- We show that VAEs unveil naturally a latent space with a structure that can be modeled as a Riemannian manifold through the learned covariance matrices in the variational posterior distributions and that such modeling can lead to better interpolations.

- We propose a natural sampling scheme consisting in sampling from a uniform distribution defined on the learned manifold and given by the Riemannian metric. We show that this procedure improves the generation process from a *vanilla* VAE significantly without complexifying the model nor the training. The proposed sampling method outperforms more advanced VAE models in terms of Frechet Inception Distance [20] and Precision and

36th Conference on Neural Information Processing Systems (NeurIPS 2022).

Recall [45] scores on four benchmark datasets. We also discuss and show that it can benefit more recent VAEs as well. An implementation is available on github.

- We show that the method appears robust to dataset size changes and outperforms even more strongly peers when only *smaller* sample sizes are considered.
- We discuss the link of the proposed metric to the *pull-back* metric.

## 2  Variational autoencoders

Considering that we are given $x \in \mathbb{R}^D$ a set of data points deriving from an unknown distribution $p(x)$, a VAE aims at inferring $p$ with a parametric model $\{p_\theta, \theta \in \Theta\}$ using a maximum likelihood estimator. A key assumption behind the VAE is to assume that the generation process involves latent variables $z$ living in a lower dimensional space such that the generative model writes

$$z \sim p(z) \quad ; \quad x \sim p_\theta(x|z),$$

where $p$ is a prior distribution over the latent variables often taken as a standard Gaussian and $p_\theta(x|z)$ is referred to as the decoder and is most of the time taken as a parametric distribution the parameters of which are estimated using neural networks. Hence, the likelihood $p_\theta$ writes:

$$p_\theta(x) = \int_{\mathcal{Z}} p_\theta(x|z)p(z)dz. \tag{1}$$

As this integral is most of the time intractable so is $p_\theta(z|x)$, the posterior distribution. Hence, Variational Inference [23] is used and a simple parametrized variational distribution $q_\phi(z|x)$ is introduced to approximate the posterior $p_\theta(z|x)$. $q_\phi(z|x)$ is referred to as the *encoder* and, in the vanilla VAE, $q_\phi$ is chosen as a multivariate Gaussian whose parameters $\mu_\phi$ and $\Sigma_\phi$ are again given by neural networks. An unbiased estimate $\hat{p}_\theta$ of the likelihood $p_\theta(x)$ can then be derived using importance sampling with $q_\phi(z|x)$ and the ELBO objective follows using Jensen's inequality:

$$\log p_\theta(x) = \log \mathbb{E}_{z \sim q_\phi}[\hat{p}_\theta] \geq \mathbb{E}_{z \sim q_\phi}[\log \hat{p}_\theta] \geq \mathbb{E}_{z \sim q_\phi} \log p_\theta(x|z) - \mathrm{KL}(q_\phi(z|x)\|p(z)) = \mathcal{L} \tag{2}$$

The ELBO is now tractable since both $q_\phi(z|x)$ and $p_\theta(x|z)$ are known and so can be optimized with respect to the *encoder* and *decoder* parameters.

*Remark* 2.1. In practice, $p_\theta(x|z)$ is chosen depending on the modeling of the input data but is often taken as a simple distribution (*e.g* fixed variance Gaussian, Bernoulli ...) and a weight $\beta$ can be applied to balance the weight of the KL term [21]. Hence, the ELBO can also be seen as a two terms objective [18]. The first one is a reconstruction term given by $p_\theta(x|z)$ while the second one is a regularizer given by the KL between the variational posterior $q_\phi$ and the prior $p$. For instance, in the case of a fixed variance Gaussian for $p_\theta(x|z)$ we have

$$\mathcal{L}_{\mathrm{REC}} = \|x - \mu_\theta(z)\|_2^2, \quad \mathcal{L}_{\mathrm{REG}} = \beta \cdot \mathrm{KL}(q_\phi(z|x)\|p(z)). \tag{3}$$

## 3  Related work

A natural way to improve the generation from VAEs consists in trying to use more complex priors [22] than the standard Gaussian distribution used in the initial version such that they better match the true distribution of the latent codes. For instance, using a Mixture of Gaussian [35, 16] or a Variational Mixture of Posterior (VAMP) [50] as priors was proposed. In the same vein, hierarchical latent variable models [48, 26] or prior learning [11, 2] have recently emerged and aimed at finding the best suited prior distribution for a given dataset. Acceptance/rejection sampling method was also proposed to try to improve the expressiveness of the prior distribution [4]. Some recent works linking energy-based models (EBM) and VAEs [51] or modeling the prior as an EBM [38] have demonstrated promising results and are also worth citing.

On the ground that the latent space must adapt to the data as well, *geometry-aware* latent space modelings as hypershpere [15], torus [17] or Poincaré disk [34] or discrete latent representations [41] were proposed. Other recent contributions proposed to see the latent space as a Riemannian manifold where the Riemannian metric is given by the Jacobian of the generator function [3, 10, 47]. This metric was then used directly within the prior modeled by Brownian motions [24]. Others

proposed to learn the metric directly from the data throughout training thanks to *geometry-aware* normalizing flows [9] or learn the latent structure of the data using transport operators [12]. While these geometry-based methods show interesting properties of the learned latent space they either require the computation of a time consuming model-dependent function, the Jacobian, or add further parameters to the model to learn the metric or transport operators adding some computational burden.

Arguing that VAEs are essentially autoencoders regularized with a Gaussian noise, Ghosh et al. [18] proposed another interesting interpretation of the VAE framework and showed that other types of regularization may be of interest as well. Since the generation process from these autoencoders is no longer relying on the prior distribution, the authors proposed to use ex-post density estimation by fitting simple distributions such as a Gaussian mixture in the latent space. While this paves the way for consideration of other ways to generate data, it mainly reduces the VAE framework to an autoencoder while we believe that it can also unveil interesting geometrical aspects.

Another widely discussed improvement of the model consists in trying to tweak the approximate posterior in the ELBO so that it better matches the true posterior using MCMC methods [46] or normalizing flows [42]. For instance, methods using Hamiltonian equations in the flows to target the true posterior [7] were proposed.

Finally, while discussing the potential link between PCA and autoencoders some intuitions arose on the impact of both the intrinsic structure of the variance of the data [40] and the shape of the covariance matrices in the posterior distributions [44] on disentanglement in the latent space. We also believe that these covariance matrices indeed play a crucial role in the modeling of the latent space but in this paper, we instead propose to see their inverse as the value of a Riemannian metric.

## 4 Proposed method

In this section, we show that a vanilla VAE unveils naturally a Riemannian structure in its latent space through the learned covariance matrices in the variational posterior distribution. We then propose a new natural generation scheme guided by this estimated geometry and consisting in sampling from a uniform distribution deriving intrinsically from the learned Riemannian manifold.

### 4.1 A word on Riemannian geometry

First, we briefly recall some basic elements of Riemannian geometry needed in the rest of the paper. A more detailed discussion on Riemannian manifolds may be found in Appendix A. A $d$-dimensional manifold $\mathcal{M}$ is a manifold which is locally homeomorphic to a $d$-dimensional Euclidean space. If the manifold $\mathcal{M}$ is further differentiable it possesses a tangent space $T_z$ at any $z \in \mathcal{M}$ composed of the tangent vectors of the curves passing by $z$. If $\mathcal{M}$ is equipped with a smooth inner product $g : z \to \langle \cdot | \cdot \rangle_z$ defined on its tangent space $T_z$ for any $z \in \mathcal{M}$ then $\mathcal{M}$ is called a Riemannian manifold and $g$ is the associated Riemannian metric. Then, a local representation of $g$ at any $z \in \mathcal{M}$ is given by the positive definite matrix $\mathbf{G}(z)$ (See Appendix A). If $\mathcal{M}$ is connected, a Riemannian distance between two points $z_1$, $z_2$ of $\mathcal{M}$ can be defined

$$\text{dist}_{\mathbf{G}}(z_1, z_2) = \inf_{\gamma} \int_a^b \sqrt{\dot{\gamma}(t)^\top \mathbf{G}(\gamma(t))\dot{\gamma}(t)}dt = \inf_{\gamma} L(\gamma) \quad \text{s.t.} \quad z_1 = \gamma(a), z_2 = \gamma(b)\,, \quad (4)$$

where $L$ is the length of curves $\gamma : \mathbb{R} \to \mathcal{M}$ traveling from $z_1$ to $z_2$. Curves minimizing $L$ and parametrized proportionally to the arc length are *geodesic*. The manifold $\mathcal{M}$ is said to be *geodesically complete* if all geodesic curves can be extended to $\mathbb{R}$. In an Euclidean space, $\mathbf{G}$ reduces to $I_d$ and the distance becomes the classic Euclidean one. A simple extension of this Euclidean framework consists in assuming that the metric is given by a constant positive definite matrix $\mathbf{\Sigma}$ different from $I_d$. In such a case the induced Riemannian distance is the well-known Mahalanobis distance $\text{dist}_{\mathbf{\Sigma}}(z_1, z_2) = \sqrt{(z_2 - z_1)^\top \mathbf{\Sigma}(z_2 - z_1)}$.

### 4.2 The Riemannian Gaussian distribution

Given the Riemannian manifold $\mathcal{M}$ endowed with the Riemannian metric $\mathbf{G}$ and a chart $z$, an infinitesimal volume element may be defined on each tangent space $T_z$ of the manifold $\mathcal{M}$ as follows

$$d\mathcal{M}_z = \sqrt{\det \mathbf{G}(z)}dz\,, \quad (5)$$

with $dz$ being the Lebesgue measure. This defines a canonical measure on the manifold and allows to extend the notion of random variables to Riemannian manifolds whose density can be defined with respect to that Riemannian measure (see Appendix A). Hence, a Riemannian Gaussian distribution on $\mathcal{M}$ can be defined using the Riemannian distance of Eq. (4) instead of the Euclidean one.

$$\mathcal{N}_{\text{riem}}(z|\sigma,\mu) = \frac{1}{C}\exp\Big(-\frac{\text{dist}_{\mathbf{G}}(z,\mu)^2}{2\sigma}\Big), \qquad C = \int_{\mathcal{M}} \exp\Big(-\frac{\text{dist}_{\mathbf{G}}(z,\mu)^2}{2\sigma}\Big)d\mathcal{M}_z, \quad (6)$$

where $d\mathcal{M}_z$ is the volume element defined in Eq. (5). Thus, a multivariate normal distribution with covariance matrix $\mathbf{\Sigma}$ is only a specific case of the Riemannian distribution with $\sigma = 1$ and defined on the manifold $\mathcal{M} = (\mathbb{R}^d, \mathbf{G})$ where $\mathbf{G}$ is the constant Riemannian metric $\mathbf{G}(z) = \mathbf{\Sigma}^{-1}$, $\forall z \in \mathcal{M}$.

### 4.3 Geometrical interpretation of the VAE framework

Within the VAE framework, the variational distribution $q_\phi(z|x)$ is often chosen as a simple multivariate Gaussian distribution defined on $\mathbb{R}^d$ with $d$ being the latent space dimension. Hence, as explained in the previous section, given an input data point $x_i$, the posterior $q_\phi(z|x_i) = \mathcal{N}(\mu(x_i), \mathbf{\Sigma}(x_i))$ can also be seen as a Riemannian Gaussian distribution where the Riemannian distance is simply the distance with respect to the metric tensor $\mathbf{\Sigma}^{-1}(x_i)$. Hence, the VAE framework can be seen as follow: As with an autoencoder, the VAE provides a code $\mu(x_i)$ which is a lower dimensional representation of an input data point $x_i$. However, it also gives a tensor $\mathbf{\Sigma}^{-1}(x_i)$ depending on $x_i$ which can be seen as the value of a Riemannian metric $\mathbf{G}$ at $\mu(x_i)$ *i.e.*

$$\mathbf{G}(\mu(x_i)) = \mathbf{\Sigma}^{-1}(x_i)\,.$$

This metric is crucial since it impacts the notion of distance in the latent space now seen as the Riemannian manifold $\mathcal{M} = (\mathbb{R}^d, \mathbf{G})$ and so changes the directions that are favored in the sampling from the posterior distribution $q_\phi(z|x_i)$. Then, a sample $z$ is drawn from a standard (*i.e.* $\sigma = 1$ in Eq. (6)) Riemannian Gaussian distribution and fed to the decoder. Since we only have access to a finite number of metric tensors $\mathbf{\Sigma}^{-1}(x_i)$, as a first approximation the VAE model assumes that the metric is locally constant close to $\mu(x_i)$ and so the Riemannian distance reduces to the Mahalanobis distance in the posterior distribution. This drastically simplifies the training process since now Riemannian distances have closed form and so are easily computable. Interestingly, the VAE framework will impose through the ELBO expression given in Eq. (3), that $z$ gives a sample $x \sim p_\theta(x|z)$ close to $x_i$ when decoded. Since $z$ has a probability density function imposing higher probability for samples having the smallest Riemannian distance to $\mu$, the VAE imposes in a way that latent variables that are close in the latent space with respect to the metric $\mathbf{G}$ will also provide samples that are close in the data space $\mathcal{X}$ in terms of L2 distance as noticed in Remark. 2.1. Noteworthy is that the latter distance can be amended through the choice of the decoder $p_\theta(x|z)$. This is an interesting property since it allows the VAE to directly link the learned Riemannian distance in the latent space to the distance in the data space. The regularization term in Eq. (3) ensures that the covariance matrices do not collapse to $\mathbf{0}_d$ and constraints the latent codes to remain close to the origin easing optimization. Finally, at the end of training, we have a lower dimensional representation of the training data given by the means of the posteriors $\mu(x_i)$ and a family of metric tensors $(\mathbf{G}_i = \mathbf{\Sigma}^{-1}(x_i))$ corresponding to the value of a Riemannian metric defined locally on the latent space. Inspired from [19], we propose to build a smooth continuous Riemannian metric defined on the entire latent space as follows:

$$\mathbf{G}(z) = \sum_{i=1}^{N} \mathbf{\Sigma}^{-1}(x_i) \cdot \omega_i(z) + \lambda \cdot e^{-\tau\|z\|_2^2} \cdot I_d\,, \quad \omega_i(z) = \exp\left(-\frac{\text{dist}_{\mathbf{\Sigma}^{-1}(x_i)}(z,\mu(x_i))^2}{\rho^2}\right), \quad (7)$$

where $\text{dist}_{\mathbf{\Sigma}^{-1}(x_i)}(z,\mu(x_i))^2 = (z - \mu(x_i))^\top \mathbf{\Sigma}^{-1}(x_i)(z - \mu(x_i))$ is the Riemannian distance between $z$ and $\mu(x_i)$ with respect to the locally constant metric $\mathbf{G}(\mu(x_i)) = \mathbf{\Sigma}^{-1}(x_i)$. Since the sum in Eq. (7) is made on the total number of training samples $N$, the number of centroids $(\mu(x_i))$ and so of reference metric tensors can be decreased for huge datasets by selecting only $k < N$ elements[1] and increasing $\rho$ to reduce memory usage. We provide an ablation study on the impact of $\lambda$, the number of centroids and their choice along with a discussion on the choice for $\rho$ in Appendix F. The parameter $\tau$ is only there to ensure that the volume of $(\mathbb{R}^d, \mathbf{G})$ is finite, a property that is needed in Sec. 4.5, and its value can be set as close as desired to zero so the norm of $z$ does not influence the metric

---

[1]This may be performed with $k$-medoids algorithm.

close to the centroids. In practice, it is set below computer precision (*i.e.* $\tau \approx 0$). Rigorously, the metric defined in Eq. (7) should have been used during the training process. Nonetheless, this would have made the training longer and trickier since it would involve i) the computation of Riemannian distances that have no longer closed form and so make the resolution of the optimization problem in Eq. (4) needed, ii) the sampling from Eq. (6) which is not trivial and iii) the computation of the regularization term. Moreover, for small values of $\beta$ in Eq. (3), the samples generated from the variational distribution $z \sim \mathcal{N}(\mu(x_i), \Sigma(x_i))$ can be assumed to be concentrated around $\mu(x_i)$ and so we have the following first-order Taylor expansion around $\mu(x_i)$

$$\mathbf{G}(z) \approx \Sigma^{-1}(x_i) + \sum_{j=1, j \neq i}^{N} \Sigma^{-1}(x_j) \cdot \underbrace{\omega_j(\mu(x_i))}_{\approx 0} + \Sigma^{-1}(x_i) \cdot \underbrace{\mathbf{J}_{\omega_i}(\mu(x_i))}_{=0}(z - \mu(x_i)), \quad (8)$$

where $\mathbf{J}_{\omega_i}(\mu(x_i))$ is the Jacobian of the interpolant $\omega_i$ evaluated at $\mu(x_i)$. Note that we have further assumed small enough $\rho$ and $\lambda$ to neglect the influence of the other $\Sigma(x_j)$ in Eq. (7). Hence by approximating the value of the metric during training by its value at $\mu(x_i)$ (*i.e.* $\Sigma^{-1}(x_i)$), the VAE training remains unchanged, stable and computationally reasonable since Riemannian Gaussians become multivariate Gaussians in $q_\phi(z|x)$ as explained before. Noteworthy is the fact that following the discussion on the role of the KL loss in the VAE framework and the experiments conducted in [18], in our vision of the VAE, the prior distribution is only seen as a regularizer though the KL term and other latent space regularization schemes may have been also envisioned. In the following, we keep the proposed vision and do not amend the training.

### 4.4 Link with the *pull-back* metric

It has been shown that a natural Riemannian metric on the latent space of generative models can be the *pull-back* metric given by $\mathbf{G}(z) = \mathbf{J}_g(z)^\top \mathbf{J}_g(z)$ [3] and induced by the decoder mapping $g : \mathbb{R}^d \to \mathbb{R}^D$ outputing the parameters of the conditional distribution $p_\theta(x|z)$. Actually, there exists a strong relation linking the metric proposed in this paper to the *pull-back* metric. Indeed, assuming that samples from the variational posterior $z \sim q_\phi(z|x) = \mathcal{N}(\mu(x), \Sigma(x))$ remain close to $\mu(x)$ (*e.g.* by setting a small $\beta$ in Eq. (3)) allows to consider an approximation of the log density $h(z) := \log p_\theta(x|z)$ next to $\mu(x)$ for a given $x$ [28].

$$h(z) \approx h(\mu(x)) + \mathbf{J}_h(\mu(x))(z - \mu(x)) + \frac{1}{2}(z - \mu(x))^\top \mathbf{H}_h(\mu(x))(z - \mu(x)),$$

where $\mathbf{J}_h(\mu(x))$ is the Jacobian and $\mathbf{H}_h(\mu(x))$ is the Hessian of h. Using this and remarking that

$$\mathbb{E}_{z \sim q_\phi}\left[\mathbf{J}_h(\mu)(z - \mu)\right] = 0 \quad \text{and} \quad \mathbb{E}_{z \sim q_\phi}\left[(z - \mu)^\top \mathbf{H}_h(\mu)(z - \mu)\right] = \mathrm{Tr}(\mathbf{H}_h(\mu)\Sigma),$$

makes the ELBO in Eq. (3) write:

$$\mathcal{L} \approx h(\mu(x)) + \frac{1}{2}\mathrm{Tr}(\mathbf{H}_h(\mu(x))\Sigma(x)) - \beta \, \mathrm{KL}(q_\phi(z|x) \| p(z)). \quad (9)$$

Assuming a standard Gaussian prior, Kumar and Poole [28] showed that $\widetilde{\Sigma}$ maximizing the ELBO is

$$\widetilde{\Sigma}(x) = \left(I_d - \frac{1}{\beta}\mathbf{H}_h(\mu(x))\right)^{-1}, \quad (10)$$

and if we further assume some regularity on the neural networks used for the decoder mapping $g$ (*e.g.* piece-wise linear activation functions) we have

$$\widetilde{\Sigma}(x) = \left(I_d - \frac{1}{\beta}\mathbf{J}_g(\mu(x))^\top \mathbf{H}_p(g(\mu(x)))\mathbf{J}_g(\mu(x))\right)^{-1}, \quad (11)$$

where $\mathbf{H}_p(g(\mu(x)))$ is the Hessian of $\log p_\theta(x; g(z))$. A standard case for the VAE is to assume that $p_\theta(x|z) = \mathcal{N}(\mu_\theta(z), \sigma \cdot I_D)$ and so gives $\mathbf{H}_p(g(\mu(x))) = -\frac{1}{\sigma} \cdot I_D$. If we further set $\sigma = \frac{1}{\beta}$, Eq. (11) gives a relation between the *pull-back* and the metric we propose

$$\widetilde{\Sigma}^{-1}(x) = \mathbf{J}_g(\mu(x))^\top \mathbf{J}_g(\mu(x)) + I_d.$$

Hence, the proposed metric is closely linked to the *pull-back* metric and may be useful to approximate it (at least close to the $\mu(x)$) and so avoid the computation of a potentially costly function.

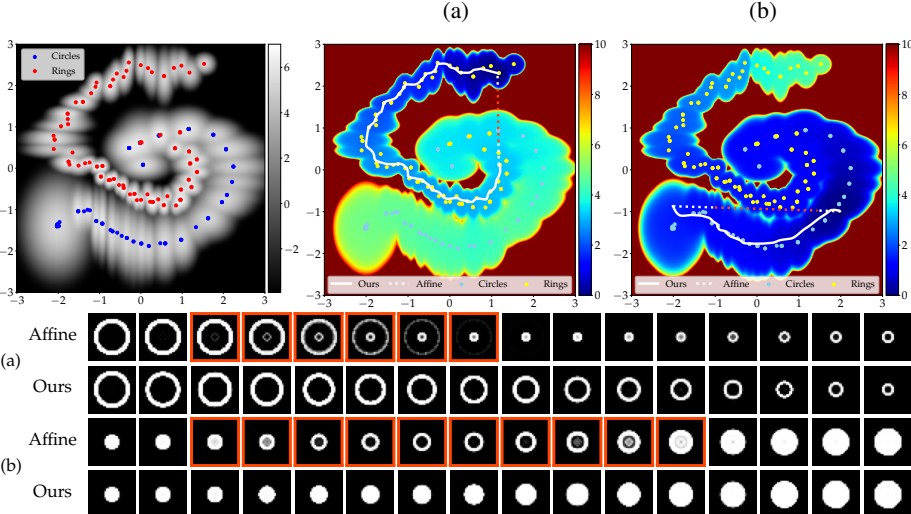

Figure 1: *Top left:* Visualization and interpolation in a 2D latent space learned by a VAE trained with binary images of rings and disks. The log of the metric volume element $\sqrt{\det \mathbf{G}(z)}$ (proportional to the log of the density we propose to sample from) is shown in gray scale. *Top middle and right*: Riemannian distance from a starting point (color maps). The dashed lines are affine interpolations between two points in the latent space and the solid ones are obtained by solving Eq. (12). *Bottom:* Decoded samples along the interpolation curves.

## 4.5 Geometry-aware sampling

Assuming that the VAE has learned a latent representation of the data in a space seen as a Riemannian manifold, we propose to exploit this strong property to enhance the generation procedure. A natural way to sample from such a latent space would consist in sampling from the uniform distribution intrinsically defined on the learned manifold. Similar to the Gaussian distribution presented in Sec. 4.2, the notion of uniform distribution can indeed be extended to Riemannian manifolds. Given a set $\mathcal{A} \subset \mathcal{M}$ having a finite volume, a Riemannian uniform distribution on $\mathcal{A}$ writes [39]

$$p_{\mathcal{A}}(z) = \frac{\mathbf{1}_{\mathcal{A}}(z)}{\mathrm{Vol}(\mathcal{A})} = \frac{\mathbf{1}_{\mathcal{A}}(z)}{\int_{\mathcal{M}} \mathbf{1}_{\mathcal{A}}(z) d\mathcal{M}_z}.$$

This density is taken with respect to $d\mathcal{M}_z$, the Riemannian measure but using Eq. (5) and a coordinate system $z$ allows to obtain a pdf defined with respect to the Lebesgue one. Moreover, since the volume of the whole manifold $\mathcal{M} = (\mathbb{R}^d, \mathbf{G})$ is finite, we can now define a *uniform distribution* on $\mathcal{M}$

$$\mathcal{U}_{\mathrm{Riem}}(z) = \frac{\sqrt{\det \mathbf{G}(z)}}{\int_{\mathbb{R}^d} \sqrt{\det \mathbf{G}(z)dz}}.$$

Since the Riemannian metric has a closed form expression given by Eq. (7) sampling from this distribution is quite easy and may be performed using the HMC sampler [36] for instance. Now we are able to sample from the intrinsic uniform distribution which is a natural way of exploring the estimated manifold and the sampling is guided by the geometry of the latent space. A discussion on practical outcomes can be found in Appendix B. Noteworthy is the fact that this approach can also be easily applied to more recent VAE models having a Gaussian posterior (*e.g.* [6, 29, 50, 32]). We detail this and show that the proposed method can also benefit these models in Appendix G.

## 4.6 Illustration on a toy dataset

The usefulness of such sampling procedure can be observed in Figure 1 where a vanilla VAE was trained with a toy dataset composed of binary images of disks and rings of different size and thickness (example inspired by [8]). On the left is presented the learned latent space along with the embedded training points given by the colored dots. The log of the metric volume element is given in gray scale.

In this example, we clearly see a geometrical structure appearing since the disks and rings seem to wrap around each other. Obviously, sampling using the prior (taken as a $\mathcal{N}(0, I_d)$) in such a case is far from being optimal since the sampling will be performed regardless of the underlying distribution of the latent variables and so will create irrelevant samples. To further illustrate this, we propose to interpolate between points in the latent space using different cost functions. Dashed lines represent affine interpolations while the solid ones show interpolations aiming at minimizing the potential $V(z) = (\sqrt{\det \mathbf{G}(z)})^{-1}$ all along the curve *i.e.* solving the minimization problem

$$\inf_{\gamma} \int_0^1 V(\gamma(t))dt \quad \text{s.t.} \quad \gamma(0) = z_1, \ \gamma(1) = z_2 \,. \tag{12}$$

In Figure 1 are presented the decoded samples all along the interpolation curves. Thanks to those interpolations we can see that i) the latent space seems to really have a specific geometrical structure since decoding all along the interpolation curves obtained by solving Eq. (12) leads to qualitatively satisfying results, ii) certain locations of the latent space must be avoided since sampling there will produce irrelevant samples (see red frames and corresponding red dashes). Using the proposed sampling scheme will allow to sample in the light-colored areas and so ensure that the sampling remains close to the data *i.e.* where information is available and so does not produce irrelevant images when decoded while still proposing relevant variations from the input data.

## 5  Experiments

In this section, we conduct a comparison with other VAE models using other regularization schemes, more complex priors, richer posteriors, ex-post density estimation or trying to take into account geometrical aspects. In the following, all the models share the same auto-encoding neural network architectures and we used the code and hyper-parameters provided by the authors if available[2]. See Appendix D for models descriptions and the comprehensive experimental set-up.

### 5.1  Generation with benchmark datasets

First, we compare the proposed sampling method to several VAE variants such as a Wasserstein Autoencoder (WAE) [49], Regularized Autoencoders (RAEs) [18], a vamp-prior VAE (VAMP) [50], a Hamiltonian VAE (HVAE) [7], a geometry-aware VAE (RHVAE) [9] and an Autoencoder (AE). We elect these models since they use different ways to generate the data using either the prior or ex-post density estimation. For the latter, we fit a 10-component mixture of Gaussian in the latent space after training like [18] .

Figure 2 shows a qualitative comparison between the resulting generated samples for MNIST [30] and CELEBA [31], see Appendix C for SVHN [37] and CIFAR 10 [27]. Interestingly, using the non-prior based methods seems to produce qualitatively better samples (rows 7 to end). Nonetheless, the resulting samples seem even sharper when the sampling takes into account geometrical aspects of the latent space as we propose (last row). Additionally, even though the exact same model is used, we clearly see that using the proposed method represents a strong improvement of the generation process from a vanilla VAE when compared to the samples coming from a normal prior (second row). This confirms that even the simplest VAE model actually contains a lot of information in its latent space but the limited expressiveness of the prior impedes to access to it. Hence, using more complex priors such as the VAMP may be a tempting idea. However, one must keep in mind that the ELBO objective in Eq. (2) must remain tractable and so using more expressive priors may be impossible.

These observations are even more supported by Table 1 where we report the Frechet Inception Distance (FID) and the precision and recall (PRD) score against the test set to assess the sampling quality and diversity. Again, fitting a mixture of Gaussian (GMM) in the latent space appears to be an interesting idea since it allows for a better expressiveness and latent space prospecting. For instance, on MNIST the FID falls from 40.7 with the prior to 13.1 when using a GMM. Nonetheless, with the proposed method we are able to make it even smaller (8.5) and PRD scores higher without changing the model and performing post processing. This can also be observed on the 3 other datasets. Impressively, in almost all cases, the proposed generation method can either compete or outperform peers both in terms of FID and PRD scores.

---

[2]We also perform a wider hyper-parameter search on MNIST and CELEBA for each model in Appendix C

|            | MNIST | CELEBA |
|------------|-------|--------|

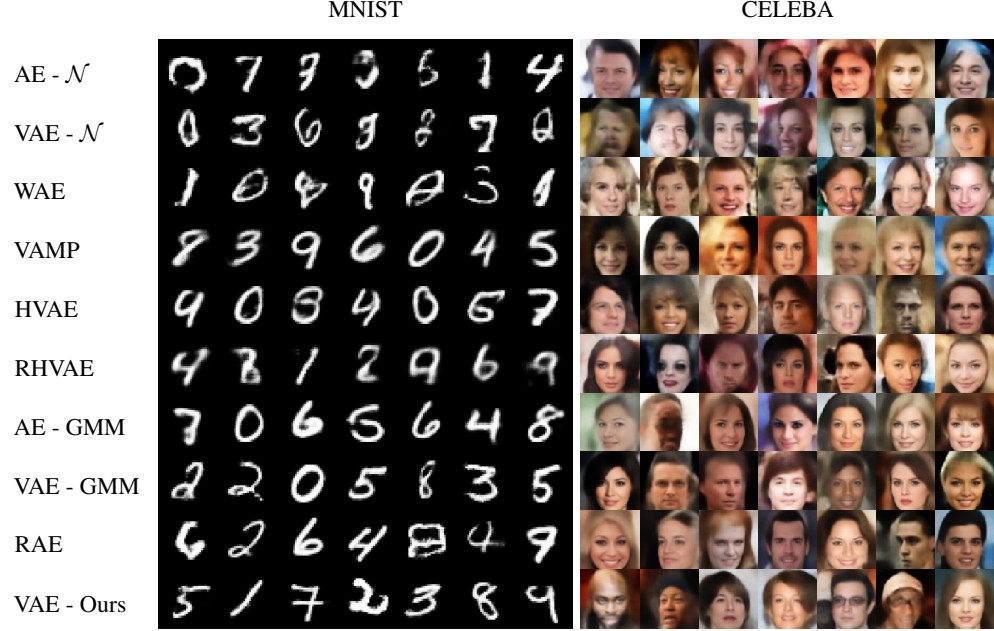

Figure 2: Generated samples with different models and generation methods. Results with RAE variants are provided in Appendix C.

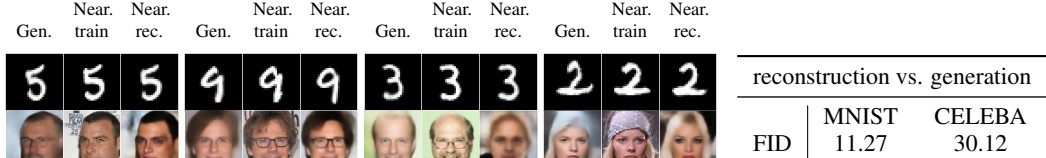

| | | reconstruction vs. generation | |
|---|---|---|---|
| | | MNIST | CELEBA |
| FID | | 11.27 | 30.12 |

Figure 3: *Left:* Nearest train image (near. train) and nearest image in all reconstructions of train images (near. rec.) to the generated one (Gen.) with the proposed method. Note: the nearest reconstruction may be different from the reconstruction of the nearest train image. *Right*: The FID score between 10k generated images and 10k reconstructed train samples.

Finally, we check if the proposed method does not overfit the training data and is able to produce diverse samples by showing the nearest neighbor in the train set and the nearest image in all the reconstructions of the train images to a generated image in Figure 3 (left). We also provide the FID score between 10k generated samples and 10k train reconstructions in Figure 3 (right). These experiments show that the generated samples are not only resampled train images and that the sampling prospects quite well the manifold. To support even more this claim we provide in Appendix F an analysis in a case where only two centroids are selected in the metric. This also shows that the generated samples are not only an interpolation between the $k$ selected centroids since some generated images contain attributes that are not present in the images of the decoded centroids.

The outcome of such an experiment is that using post training latent space processing such as ex-post density estimation or adding some geometrical consideration to the model allows to strongly improve the sampling without adding more complexity to the model. Generating 1k samples on CELEBA takes approx. 5.5 min for our method vs. 4 min for a 10-component GMM on a GPU V100-16GB.

## 5.2 Investigating robustness in low data regime

We perform a comparison using the same models and datasets as before but we progressively decrease the size of the training set to see the robustness of the methods according to the number of samples. Despite rarely performed in most generative models related papers, this setup appeared to us relevant

Table 1: FID (lower is better) and PRD score (higher is better) for different models and datasets. For the mixture of Gaussian (GMM), we fit a 10-component mixture of Gaussian in the latent space.

| Model | MNIST (16) | | SVHN (16) | | CIFAR 10 (32) | | CELEBA (64) | |
|---|---|---|---|---|---|---|---|---|
| | FID $\downarrow$ | PRD $\uparrow$ | FID $\downarrow$ | PRD $\uparrow$ | FID $\downarrow$ | PRD $\uparrow$ | FID $\downarrow$ | PRD $\uparrow$ |
| AE - $\mathcal{N}(0,1)$ | 46.41 | 0.86/0.77 | 119.65 | 0.54/0.37 | 196.50 | 0.05/0.17 | 64.64 | 0.29/0.42 |
| WAE | 20.71 | 0.93/0.88 | 49.07 | 0.80/**0.85** | 132.99 | 0.24/0.52 | 54.56 | **0.57**/0.55 |
| VAE - $\mathcal{N}(0,1)$ | 40.70 | 0.83/0.75 | 83.55 | 0.69/0.55 | 162.58 | 0.10/0.32 | 64.13 | 0.27/0.39 |
| VAMP | 34.02 | 0.83/0.88 | 91.98 | 0.55/0.63 | 198.14 | 0.05/0.11 | 73.87 | 0.09/0.10 |
| HVAE | 15.54 | 0.97/0.95 | 98.05 | 0.64/0.68 | 201.70 | 0.13/0.21 | 52.00 | 0.38/0.58 |
| RHVAE | 36.51 | 0.73/0.28 | 121.69 | 0.55/0.41 | 167.41 | 0.12/0.22 | 55.12 | 0.45/0.56 |
| AE - GMM | 9.60 | 0.95/0.90 | 54.21 | 0.82/0.83 | 130.28 | 0.35/0.58 | 56.07 | 0.32/0.48 |
| RAE (GP) | 9.44 | 0.97/**0.98** | 61.43 | 0.79/0.78 | 120.32 | 0.34/0.58 | 59.41 | 0.28/0.49 |
| RAE (L2) | 9.89 | 0.97/**0.98** | 58.32 | 0.82/0.79 | 123.25 | 0.33/0.54 | 54.45 | 0.35/0.55 |
| RAE (SN) | 11.22 | 0.97/**0.98** | 95.64 | 0.53/0.63 | 114.59 | 0.32/0.53 | 55.04 | 0.36/0.56 |
| RAE | 11.23 | **0.98/0.98** | 66.20 | 0.76/0.80 | 118.25 | 0.35/0.57 | 53.29 | 0.36/0.58 |
| VAE - GMM | 13.13 | 0.95/0.92 | 52.32 | 0.82/**0.85** | 138.25 | 0.29/0.53 | 55.50 | 0.37/0.49 |
| VAE - Ours | **8.53** | **0.98**/0.97 | **46.99** | **0.84/0.85** | **93.53** | **0.71/0.68** | **48.71** | 0.44/**0.62** |

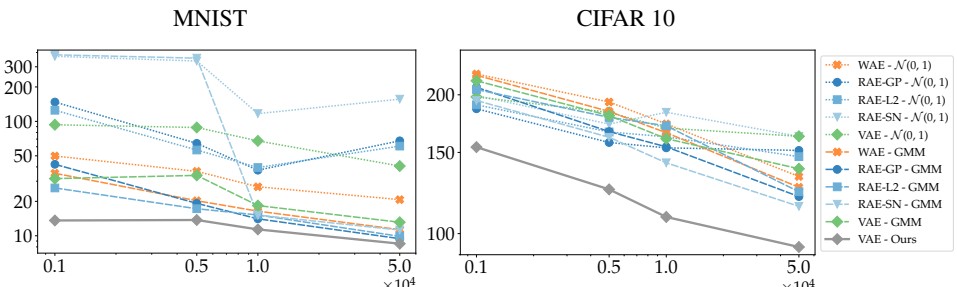

Figure 4: Evolution of the FID score according to the number of training samples.

since 1) it is well known that it may be challenging for these models, 2) in day-to-day applications collecting large databases may reveal costly if not impossible (*e.g.* medicine). Hence, we consider MNIST, CIFAR10 and SVHN and use either the full dataset size, 10k, 5k or 1k training samples. For each experiment, the best retained model is again the one achieving the best ELBO on the validation set the size of which is set as 20% of the train size. See Appendix D for further details about experiments set-up. Then, we report the evolution of the FID against the test set in Figure 4. Results obtained on SVHN are presented in Appendix E. Again, the proposed sampling method appears quite robust to the dataset size since it outperforms the other models' FID even when the number of training samples is smaller. This is made possible thanks to the proposed metric that allows to avoid regions of the latent space having poor information. Finally, our study shows that although using more complex generation procedures such as ex-post density estimation seems to still enhance the generation capability of the model when the number of training samples remains quite high ($\geq$5k), this gain seems to worsen when the dataset size reduces as illustrated on CIFAR. In addition, we also evaluate the model on a data augmentation task with neuroimaging data from OASIS [33] mimicking a day-to-day scenario where the limited data regime is very common in Appendix C.3.

## 6 Conclusion

In this paper, we provided a geometric understanding of the latent space learned by a VAE and showed that it can actually be seen as a Riemannian manifold. We proposed a new natural generation process consisting in sampling from the intrinsic uniform distribution defined on this learned manifold. The proposed method was empirically shown to be competitive with more advanced versions of the VAEs using either more complex priors, ex-post density estimation, normalizing flows or other regularization schemes. Interestingly, the proposed method revealed good robustness properties in complex settings such as high dimensional data or low sample sizes and appeared to benefit more recent VAE models as well. Future work would consist in trying to use this method to perform data augmentation in those challenging contexts and compare its reliability for such a task with state of the art methods or trying to use this metric to perform clustering in the latent space.

## Acknowledgments and Disclosure of Funding

The research leading to these results has received funding from the French government under management of Agence Nationale de la Recherche as part of the "Investissements d'avenir" program, reference ANR-19-P3IA-0001 (PRAIRIE 3IA Institute) and reference ANR-10-IAIHU-06 (Agence Nationale de la Recherche-10-IA Institut Hospitalo-Universitaire-6). This work was granted access to the HPC resources of IDRIS under the allocation AD011013517 made by GENCI (Grand Equipement National de Calcul Intensif).

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
