# A Further elements on Riemannian geometry

A $d$-dimensional Riemannian manifold $\mathcal{M}$ can be defined as a $d$-dimensional differentiable manifold equipped with is a smooth inner product $g : z \to \langle \cdot | \cdot \rangle_z$ defined on each tangent space $T_z\mathcal{M}$ of the manifold with $z \in \mathcal{M}$. A chart (or coordinate system) $(U, \phi)$ is a homeomorphism mapping an open set $U$ of the manifold to an open set $V$ of an Euclidean space. Given $z \in U$, a chart $\phi : (z^1, \ldots, z^d)$ induces a basis $\left( \frac{\partial}{\partial z^1}, \ldots, \frac{\partial}{\partial z^d} \right)_z$ on the tangent space $T_z\mathcal{M}$. Hence, the metric of a Riemannian manifold can be locally represented in the chart $\phi$ as a positive definite matrix as mentionned in Sec. 4.1.

$$\mathbf{G}(z) = (g_{i,j})_{z,0 \leq i,j \leq d} = \left( \left\langle \frac{\partial}{\partial z^i} \Big| \frac{\partial}{\partial z^j} \right\rangle_z \right)_{0 \leq i,j \leq d}, \tag{1}$$

for each point $z$ of the manifold. That is for $v, w \in T_z\mathcal{M}$ and $z \in \mathcal{M}$, the inner product writes $\langle u | w \rangle_z = u^\top \mathbf{G}(z) w$. Assuming that the manifold is also connected, for any $z_1, z_2 \in \mathcal{M}$, two points of the manifold, we can consider a curve $\gamma$ traveling in $\mathcal{M}$ and parametrized by $t \in [a, b]$ such that $\gamma(a) = z_1$ and $\gamma(b) = z_2$. Then, the length of $\gamma$ is given by

$$L(\gamma) = \int_a^b \|\dot{\gamma}(t)\|_{\gamma(t)} dt = \int_a^b \sqrt{\langle \dot{\gamma}(t) | \dot{\gamma}(t) \rangle_{\gamma(t)}} dt$$

Curves $\gamma$ that minimize $L$ and are parameterized proportionally to the arc length are called *geodesic* curves. A distance $\mathrm{dist}_\mathbf{G}$ on the manifold $\mathcal{M}$ can then be derived and writes

$$\mathrm{dist}_\mathbf{G}(z_1, z_2) = \inf_\gamma L(\gamma) \quad \text{s.t.} \quad \gamma(a) = z_1, \gamma(b) = z_2 \tag{2}$$

The manifold $\mathcal{M}$ is said to be *geodesically complete* if all geodesic curves can be extended to $\mathbb{R}$. Given the Riemannian manifold $\mathcal{M}$ endowed with the Riemannian metric $\mathbf{G}$ and a chart $z$, an infinitesimal volume element may also be defined on each tangent space $T_z$ of the manifold $\mathcal{M}$ as follows

$$d\mathcal{M}_z = \sqrt{\det \mathbf{G}(z)} dz, \tag{3}$$

with $dz$ being the Lebesgue measure. This defines a canonical measure on the manifold and allows to extend the notion of probability distributions to Riemannian manifolds. In particular, such a property allows to refer to random variables with a density defined with respect to the measure on the manifold. We recall such definition from [16] below

**Definition A.1.** Let $\mathcal{B}(\mathcal{M})$ be the Borel $\sigma$-algebra of $\mathcal{M}$. The random point $\mathbf{z}$ has a probability density function $\rho_\mathbf{z}$ if:

$$\forall \mathcal{Z} \in \mathcal{B}(\mathcal{M}), \ \ \mathbb{P}(\mathbf{z} \in \mathcal{Z}) = \int_\mathcal{Z} \rho(z) d\mathcal{M}(z) \ \text{ and } \int_\mathcal{M} \rho(z) d\mathcal{M}(z) = 1$$

Finally, given a chart $\phi$ defined on the whole manifold $\mathcal{M}$ and a random point $\mathbf{z}$ on $\mathcal{M}$, the point $\mathbf{p} = \phi(\mathbf{z})$ is a random point whose density $\rho'_\mathbf{p}$ may be written with respect to the Lebesgue measure as such [16]:

$$\rho'_\mathbf{p}(p) = \rho_\mathbf{z}(\phi^{-1}(p)) \sqrt{\det g(\phi^{-1}(p))} \tag{4}$$

## B The generation process algorithm - Implementation details

In this appendix, we provide pseudo-code algorithms explaining how to build the metric from a trained VAE and how to use the proposed sampling process. Noteworthy is the fact that we do not amend the training process of the vanilla VAE which remains pretty simple and stable.

### B.1 Building the metric

In this section, we explain how to build the proposed Riemannian metric. For the sake of clarity, we recall the expression of the metric below

$$\mathbf{G}(z) = \sum_{i=1}^{N} \mathbf{\Sigma}^{-1}(x_i) \cdot \omega_i(z) + \lambda \cdot e^{-\tau \|z\|_2^2} \cdot I_d \,, \tag{5}$$

where

$$\omega_i(z) = \exp\left(-\frac{\text{dist}_{\mathbf{\Sigma}^{-1}(x_i)}(z, \mu(x_i))^2}{\rho^2}\right) = \exp\left(-\frac{(z - \mu(x_i))^\top \mathbf{\Sigma}^{-1}(x_i)(z - \mu(x_i))}{\rho^2}\right),$$

---

**Algorithm 1** Building the metric from a trained model

---

**Input:** A trained VAE model $m$, the training dataset $\mathcal{X}, \lambda, \tau$        ▷ In practice $\tau \approx 0$
**for** $x_i \in \mathcal{X}$ **do**
    $\mu_i, \mathbf{\Sigma}_i = m(x_i)$        ▷ Retrieve training embeddings and covariance matrices
**end for**
Select $k$ centroids $c_i$ in the $\mu_i$        ▷ e.g. with $k$-medoids
Get corresponding covariance matrices $\mathbf{\Sigma}_i$
$\rho \leftarrow \max_i \min_{j \neq i} \|c_i - c_j\|_2$        ▷ Set $\rho$ to the max distance between two closest neighbors
Build the metric using Eq. (5)

$$\mathbf{G}(z) = \sum_{i=1}^{N} \mathbf{\Sigma}_i^{-1} \cdot \omega_i(z) + \lambda \cdot e^{-\tau \|z\|_2^2} \cdot I_d$$

**Return G**        ▷ Return **G** as a function

---

As is standard in VAE implementations, we assume that the covariance matrices $\mathbf{\Sigma}_i$ given by the VAE are diagonal and that the encoder outputs a mean vector and the log of the diagonal coefficients. In the implementation, the exponential is then applied to recover the $\mathbf{\Sigma}_i$ so that no singular matrix arises.

### B.2 Sampling process

Further to the description performed in the paper, we provide here a detailed algorithm stating the main steps of the generation process.

#### B.2.1 The HMC sampler

In the sampling process we propose to rely on the Hamiltonian Monte Carlo sampler to sample from the Riemanian uniform distribution. In a nutshell, the HMC sampler aims at sampling from a target distribution $p_{\text{target}}(z)$ with $z \in \mathbb{R}^d$ using Hamiltonian dynamics. The main idea behind such a sampler is to introduce an auxiliary random variable $v \sim \mathcal{N}(0, I_d)$ independent from $z$ and mimic the behavior of a particle having $z$ (resp. $v$) as location (resp. velocity). The Hamiltonian of the particle then writes

$$H(z, v) = U(z) + K(v) \,,$$

where $U(z)$ is the potential energy and $K(v)$ is its kinetic energy both given by

$$U(z) = -\log p_{\text{target}}(z), \qquad K(v) = \frac{1}{2} v^\top v$$

The following Hamilton's equations govern the evolution in time of the particle.

$$\begin{cases} \frac{\partial H(z,v)}{\partial v} &= v\,, \\ \frac{\partial H(z,v)}{\partial z} &= -\nabla_z \log p_{\text{target}}(z)\,. \end{cases} \tag{6}$$

In order to integrate these equations, recourse to the leapfrog integrator is needed and consists in applying $n_{\text{lf}}$ times the following equations.

$$\begin{cases} v(t + \frac{\varepsilon_{\text{lf}}}{2}) &= v(t) + \frac{\varepsilon_{\text{lf}}}{2} \cdot \nabla_z \log p_{\text{target}}(z(t))\,, \\ z(t + \varepsilon_{\text{lf}}) &= z(t) + \varepsilon_{\text{lf}} \cdot v(t + \frac{\varepsilon_{\text{lf}}}{2})\,, \\ v(t + \varepsilon_{\text{lf}}) &= v(t + \frac{\varepsilon_{\text{lf}}}{2}) + \frac{\varepsilon_{\text{lf}}}{2} \cdot \nabla_z \log p_{\text{target}}(z(t + \varepsilon_{\text{lf}}))\,, \end{cases} \tag{7}$$

where $\varepsilon_{\text{lf}}$ is called the leapfrog step size. This algorithm produces a proposal $(\widetilde{z}, \widetilde{v})$ that is accepted with probability $\alpha$ where

$$\alpha = \min\left(1, \exp\left(H(z,v) - H(\widetilde{z}, \widetilde{v})\right)\right)\,.$$

This procedure is then repeated to create an ergodic Markov chain $(z^n)$ converging to the distribution $p_{\text{target}}$ [6, 12, 15, 8].

## B.3 The proposed algorithm

In our setting the target density is given by the density of the Riemannian uniform distribution which writes with respect to Lebesgue measure as follows

$$p(z) = \mathcal{U}_{\text{Riem}}(z) = \frac{1}{C}\sqrt{\det \mathbf{G}(z)} \qquad C = \int_{\mathbb{R}^d} \sqrt{\det \mathbf{G}(z)}dz\,. \tag{8}$$

Note that thanks to the shape of the metric, this distribution is well defined since $C < +\infty$. The log density follows

$$\log p(z) = \frac{1}{2}\log \det \mathbf{G}(z) - \log C\,,$$

Hence, the Hamiltonian writes

$$H(z,v) = -\log p(z) + \frac{1}{2}v^{\top}v\,,$$

and Hamilton's equations become

$$\begin{cases} \frac{\partial H(z,v)}{\partial v} &= v\,, \\ \frac{\partial H(z,v)}{\partial z_i} &= -\frac{\partial \log p(z)}{\partial z_i} = -\frac{1}{2}\text{tr}\left(\mathbf{G}^{-1}(z)\frac{\partial \mathbf{G}(z)}{\partial z_i}\right) \end{cases}$$

Since the covariance matrices are supposed to be diagonal as is standard in VAE implementations, the computation of the inverse metric is straightforward. Moreover, since $\mathbf{G}(z)$ is smooth and has a closed form, it can be differentiated with respect to $z$ pretty easily. Now, the leapfrog integrator given in Eq. (7) can be used and the acceptance ratio $\alpha$ is easy to compute. Noteworthy is the fact that the normalizing constant $C$ is never needed since it vanishes in the gradient computation and simplifies in the acceptance ratio $\alpha$. We provide a pseudo-code of the proposed sampling procedure in Alg. 2. A typical choice in the sampler's hyper-parameters used in the paper is $N = 100$, $n_{\text{lf}} = 10$ and $\varepsilon_{\text{lf}} = 0.01$. The initialization of the chain can be done either randomly or on points that belong to the manifold (i.e. the centroids $c_i$ or $\mu(x_i)$).

---

**Algorithm 2** Proposed sampling process

---

**Input:** The metric function $\mathbf{G}$, hyper-parameters of the HMC sampler (chain length $N$, number of leapfrog steps $n_{\mathrm{lf}}$, leapfrog step size $\varepsilon_{\mathrm{lf}}$)

**Initialization:** $z$                                                       ▷ Initialize the chain

**for** $i = 1 \rightarrow N$ **do**

    $v \sim \mathcal{N}(0, I_d)$                                              ▷ Draw a velocity

    $H_0 \leftarrow H(z, v)$                              ▷ Compute the starting Hamiltonian

    $z_0 \leftarrow z$

    **for** $k = 1 \leftarrow n_{\mathrm{lf}}$ **do**

        $\bar{v} \leftarrow v - \frac{\varepsilon_{\mathrm{lf}}}{2} \cdot \nabla_z H(z, v)$

        $\widetilde{z} \leftarrow z + \varepsilon_{\mathrm{lf}} \cdot \bar{v}$                           ▷ Leapfrog step Eq. (7)

        $\widetilde{v} \leftarrow \bar{v} - \frac{\varepsilon_{\mathrm{lf}}}{2} \cdot \nabla_z H(\widetilde{z}, \bar{v})$

        $v \leftarrow \widetilde{v}$

        $z \leftarrow \widetilde{z}$

    **end for**

    $H \leftarrow H(\widetilde{z}, \widetilde{v})$                         ▷ Compute the ending Hamiltonian

    Accept $\widetilde{z}$ with probability $\alpha = \min\left(1, \exp(H_0 - H)\right)$

    **if** Accepted **then**

        $z \leftarrow \widetilde{z}$

    **else**

        $z \leftarrow z_0$

    **end if**

**end for**

**Return** $z$

---

# C   Other generation results

## C.1   Some further samples on CELEBA and MNIST

In this section, we provide some further generated samples using the proposed method. Figure 1 and Figure 2 again support the fact that the method is able to generate sharp and diverse samples. We also add the other variants of the RAE model in Figure 3.

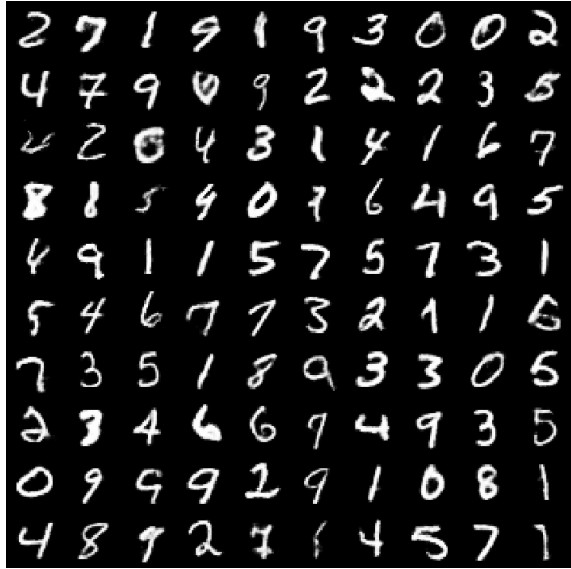

Figure 1: 100 samples with the proposed method on MNIST dataset.

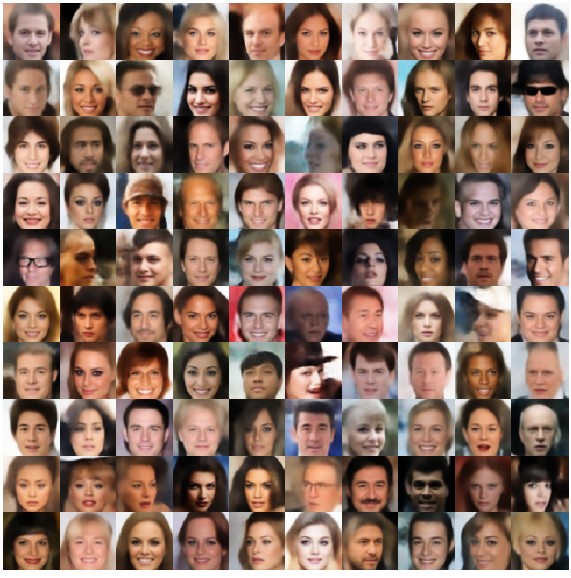

Figure 2: 100 samples with the proposed method on CELEBA dataset.

MNIST CELEBA

AE - $\mathcal{N}$

VAE - $\mathcal{N}$

WAE

VAMP

HVAE

RHVAE

AE - GMM

VAE - GMM

RAE (GP)

RAE (L2)

RAE (SN)

RAE

VAE - Ours

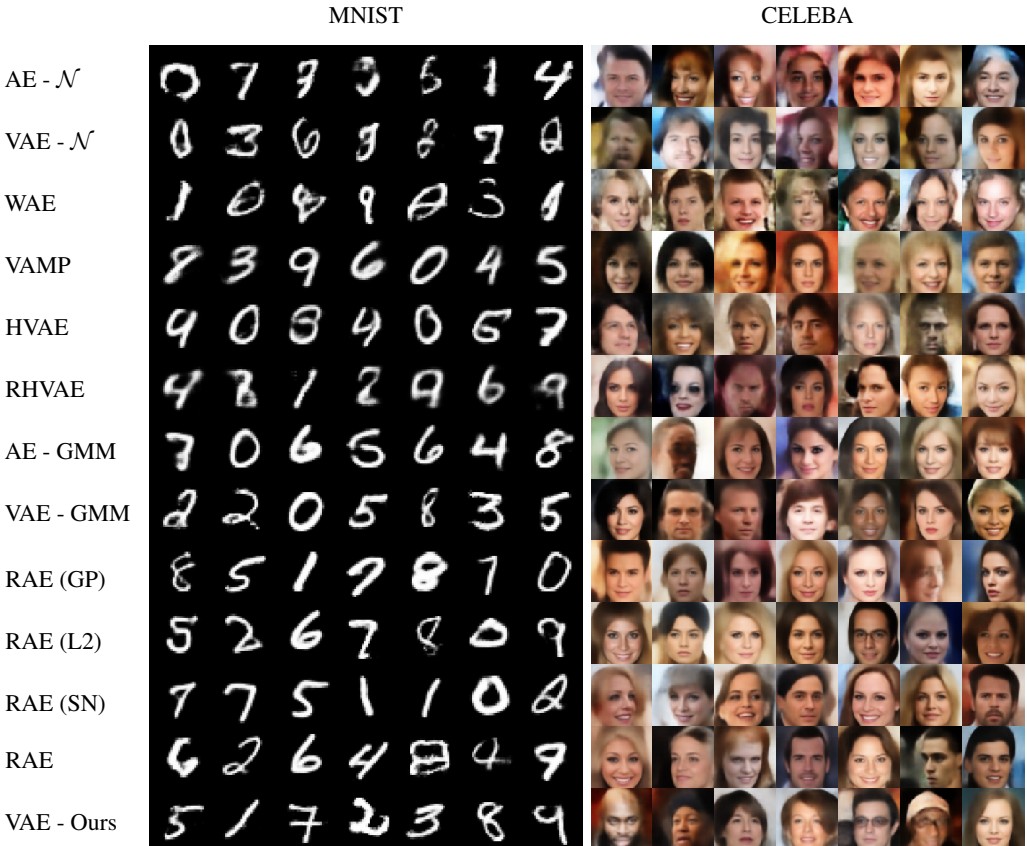

Figure 3: Generated samples with different models and generation processes.

## C.2  CIFAR and SVHN

In this appendix, we gather the resulting samplings from the different models considered for SVHN and CIFAR 10.

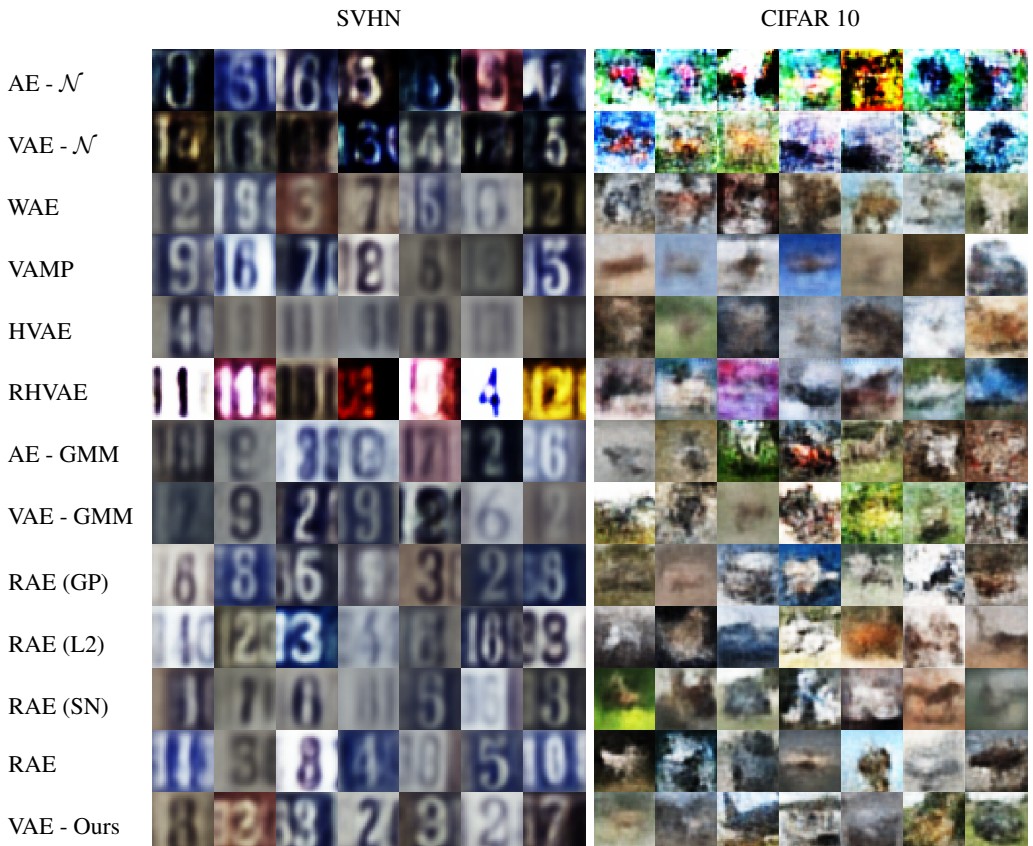

Figure 4: Generated samples with different models and generation processes.

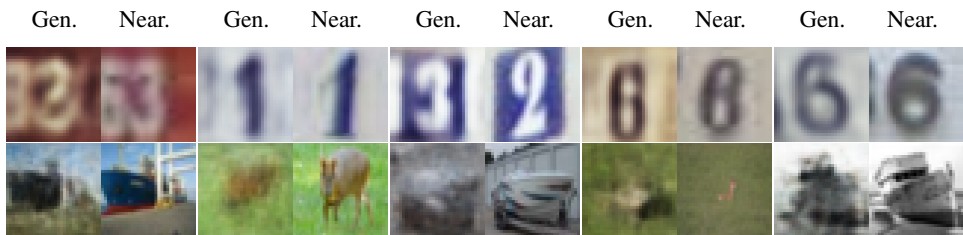

Figure 5: Closest element in the training set (Near.) to the generated one (Gen.) with the proposed method.

 **C.3    Generation with complex data**

Finally, we also propose to stress the proposed generation procedure in a day-to-day scenario where the limited data regime is more than common. To stress the model in such condition, we consider the publicly available OASIS database [14] composed of 416 MRI of patients, 100 of whom were diagnosed with Alzheimer disease (AD). Since both FID and PRD scores are not reliable when no large test set is available, we propose to assess quantitatively the generation quality with a data augmentation task. Hence, we split the dataset into a train (70%), a validation (10%) and a test set (20%). Each model is trained on each label of the train set and used to generate 2k samples per class. Then a CNN classifier is trained on i) the original train set and ii) the 4k generated samples from the generative models and tested on the test set. Table 1 shows classification results averaged across 20 runs for each considered model. The line *raw (resampled)* corresponds to a case where the train set is obtained by balancing the classes with simple repetitions of the samples from the under-represented class. These metrics provide a way to assess i) if the model can generate data adding relevant information for classification and ii) allows to quantify the amount of overfitting. The proposed method is the only one allowing to achieve higher balanced accuracy and F1 scores for both labels than on the original (unbalanced) data meaning that the samples are relevant to the classifier and this is also sign of a good generalization. Moreover, we provide generated samples using each generation procedure in Figure 6. Again, the proposed method appears to produce visually the sharpest samples. However, such augmentation method for medical data requires caution and needs further assessment on the possibly induced biases before being used on a *real-life* application case.

Table 1: Classification results averaged on 20 independent runs. For the VAEs, the classifier is trained on 2K generated samples per class.

| Generation method | Balanced Accuracy | F1 | | Precision | | Recall | |
|---|---|---|---|---|---|---|---|
| | | AD | CN | AD | CN | AD | CN |
| Original* | $66.2 \pm 7.6$ | $47.6 \pm 15.8$ | $87.3 \pm 2.0$ | $74.7 \pm 8.4$ | $80.3 \pm 4.0$ | $35.7 \pm 16.3$ | $95.7 \pm 1.5$ |
| Original (resampled) | $81.8 \pm 2.6$ | $72.1 \pm 3.6$ | $\mathbf{88.0 \pm 2.3}$ | $67.0 \pm 5.3$ | $91.4 \pm 1.8$ | $78.5 \pm 5.2$ | $85.1 \pm 4.2$ |
| AE - $\mathcal{N}$ | $50.0 \pm 0.0$ | $0.0 \pm 0.0$ | $84.1 \pm 0.0$ | $0.0 \pm 0.0$ | $72.6 \pm 0.0$ | $0.0 \pm 0.0$ | $100.0 \pm 0.0$ |
| WAE | $57.4 \pm 9.7$ | $21.0 \pm 24.5$ | $84.4 \pm 2.3$ | $48.5 \pm 42.8$ | $76.7 \pm 6.1$ | $19.3 \pm 27.5$ | $95.4 \pm 9.3$ |
| VAE - $\mathcal{N}$ | $51.8 \pm 3.8$ | $6.1 \pm 11.8$ | $84.6 \pm 1.1$ | $38.0 \pm 47.3$ | $73.4 \pm 1.7$ | $3.7 \pm 7.8$ | $99.8 \pm 0.7$ |
| VAMP | $83.1 \pm 2.6$ | $70.4 \pm 3.6$ | $82.2 \pm 4.7$ | $56.3 \pm 5.2$ | $97.5 \pm 2.1$ | $94.8 \pm 4.7$ | $71.5 \pm 7.4$ |
| HVAE | $56.3 \pm 7.9$ | $19.6 \pm 21.7$ | $85.4 \pm 1.7$ | $48.7 \pm 41.7$ | $75.5 \pm 3.8$ | $13.9 \pm 17.6$ | $98.6 \pm 2.2$ |
| RHVAE | $68.0 \pm 10.9$ | $47.0 \pm 24.2$ | $85.1 \pm 3.3$ | $56.1 \pm 25.3$ | $83.0 \pm 7.5$ | $46.7 \pm 30.2$ | $89.2 \pm 10.6$ |
| AE - GMM | $82.4 \pm 2.3$ | $69.5 \pm 3.1$ | $82.0 \pm 3.6$ | $55.8 \pm 4.9$ | $96.8 \pm 2.4$ | $93.3 \pm 5.6$ | $71.5 \pm 6.2$ |
| RAE (GP) | $63.9 \pm 9.8$ | $46.5 \pm 15.9$ | $70.6 \pm 19.6$ | $45.3 \pm 18.5$ | $84.2 \pm 8.6$ | $60.9 \pm 28.6$ | $67.0 \pm 24.9$ |
| RAE (L2) | $74.1 \pm 6.0$ | $60.6 \pm 9.5$ | $82.1 \pm 5.9$ | $57.8 \pm 10.1$ | $88.3 \pm 5.2$ | $70.0 \pm 18.7$ | $78.3 \pm 11.7$ |
| RAE (SN) | $62.3 \pm 8.9$ | $37.8 \pm 22.6$ | $80.1 \pm 7.9$ | $43.1 \pm 24.9$ | $80.6 \pm 6.6$ | $41.7 \pm 30.1$ | $82.9 \pm 16.4$ |
| RAE | $69.3 \pm 8.1$ | $53.8 \pm 12.9$ | $80.0 \pm 10.7$ | $56.2 \pm 13.5$ | $85.2 \pm 6.2$ | $60.0 \pm 24.0$ | $78.5 \pm 17.5$ |
| VAE - GMM | $83.0 \pm 3.6$ | $71.4 \pm 4.3$ | $85.3 \pm 3.0$ | $60.7 \pm 5.4$ | $94.9 \pm 3.7$ | $88.0 \pm 9.5$ | $77.9 \pm 5.9$ |
| VAE - Ours | $\mathbf{85.4 \pm 2.5}$ | $\mathbf{74.7 \pm 3.5}$ | $87.3 \pm 2.7$ | $64.0 \pm 5.3$ | $95.8 \pm 2.2$ | $90.4 \pm 5.6$ | $80.3 \pm 5.1$ |

*unbalanced

**C.4    Wider hyper-parameter search**

As stated in the paper, for the experiments, we used the official implementation and hyper-parameters provided by the authors when available. However, we also propose to perform a hyper-parameter search for the models considered in the benchmark *i.e.* WAE, VAMP-VAE, RAE-GP and RAE-L2 [3] on MNIST and CELEBA. Since both HVAE and RHVAE models have a very time consuming training, we propose to replace these approaches with models having the same objective (i.e. enriching the posterior distribution). Do to so we consider a VAE with inverse autoregressive flows [10] (VAE-IAF) and a VAE with normalizing flows with radial/planar invertible transformations [17] (VAE-NF).

We train these models with 10 different hyper-parameter configurations on MNIST and CELEBA. For the WAE, we vary the kernel bandwidth in $\{0.01, 0.1, 0.5, 1, 2, 5\}$ and change the regularization factor weighting the reconstruction and regularization in $\{0.01, 0.1, 1, 10, 100\}$. For the RAEs, we vary the L2 latent code regularization factor and the factor before the explicit regularization in $\{1e^{-6}, 1e^{-4}, 1e^{-3}, 0.01, 0.1, 1\}$. For the VAMP we vary the number of pseudo-inputs in $\{10, 20, 30, 50, 100, 150, 200, 250, 300, 189\ 500\}$. Finally, for the flow-based VAEs we vary the complexity of

the flows with different number of IAF blocks (VAE-IAF) or different flow lengths (VAE-NF). To
assess the influence of the neural architecture, the experiment is performed twice each time with a
different neural network architecture (CNN in Table. 3 or a simpler ResNet). In Table. 2, we show
the generation vs. test FID of the model achieving the lowest FID on the validation set.

Table 2: FID (lower is better) for different models and datasets. For the mixture of Gaussian (GMM),
we fit a 10-component mixture of Gaussian in the latent space.

| MODELS | MNIST | | CELEBA | |
|---|---|---|---|---|
| NETS | CNN | RESNET | CNN | RESNET |
| AE - N(0,1) | 46.4 | 221.8 | 64.6 | 275.0 |
| WAE | 18.9 | 20.3 | 54.6 | 67.1 |
| VAE - N(0,1) | 40.7 | 47.8 | 64.1 | 69.5 |
| VAMP | 34.0 | 34.5 | 56.0 | 67.2 |
| VAE-NF | 29.3 | 32.5 | 55.4 | 67.1 |
| VAE-IAF | 27.5 | 30.6 | 56.5 | 66.2 |
| AE - GMM | 9.6 | 11.0 | 56.1 | 57.4 |
| RAE-GP | 9.4 | 11.4 | 52.5 | 59.0 |
| RAE-L2 | 9.1 | 11.5 | 54.5 | 58.3 |
| VAE - GMM | 13.1 | 12.4 | 55.5 | 59.9 |
| OURS | **8.5** | **10.7** | **48.7** | **53.2** |

OASIS

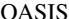

Train

VAE - $\mathcal{N}$

WAE

VAMP

HVAE

RHVAE

VAE - GMM

RAE (GP)

RAE (L2)

RAE (SN)

RAE

VAE - Ours

Figure 6: Generated samples with different models and generation processes.

## D  Experimental set-up

We compare the proposed sampling method to several VAE variants such as a Wasserstein Autoencoder (WAE) [18], Regularized Autoencoders [7] with either L2 decoder's parameters regularization (RAE-L2), gradient penalty (RAE-GP), spectral normalization (RAE-SN) or simple L2 latent code regularization (RAE), a vamp-prior VAE (VAMP) [19], a Hamiltonian VAE (HVAE) [2], a geometry-aware VAE (RHVAE) [3] and an Autoencoder (AE). The RAEs, VAEs and AEs are trained for 100 epochs for SVHN, MNIST[1] and CELEBA and 200 on CIFAR10. Each time we use the official train and test split of the data. For MNIST and SVHN, 10k samples out of the train set are reserved for validation and 40k for CIFAR10. As to CELEBA, we use the official validation set for validation. The model that is kept at the end of training is the one achieving the best validation loss. All the models are trained with a batch size of 100 and starting learning rate of $1e-3$ (but CIFAR where the learning rate is set to $5e-4$) with an Adam optimizer [9]. We also use a scheduler decreasing the learning rate by half if the validation loss stops increasing for 5 epochs. For the experiments on the sensitivity to the training set size, we keep the same set-up. For each dataset we ensure that the validation set is $1/5^{\text{th}}$ the size of the train set but for CIFAR where we select the best model on the train set. The neural networks architectures can be found in Table 3 and are inspired by [7]. The metrics (FID and PRD scores) are computed with 10000 samples against the test set (for CELEBA we selected only the 10000 first samples of the official test set). The factor $\rho$ is set to $\rho = \max\limits_{i} \min\limits_{j \neq i} \|c_i - c_j\|_2$ to ensure some *smoothness* of the manifold. For models coming from peers, we use the parameters and code provided by the authors when available and allowed by licenses.

For the data augmentation task, the generative models are trained on each class for 1000 epochs with a batch size of 100 and a starting learning rate of $1e-4$. Again a scheduler is used and the learning rate is cut by half if the loss does not improve for 20 epochs. All the models have the autoencoding architecture described in Table 3. As to the classifier, it is trained with a batch size of 200 for 50 epochs with a starting learning rate of $1e-4$ and Adam optimizer. A scheduler reducing the learning rate by half every 5 epochs if the validation loss does not improve is again used. The best kept model is the one achieving the best balanced accuracy on the validation set. Its neural network architecture may be found in Table 4. MRIs are only pre-processed such that the maximum value of a voxel is 1 and the minimum 0 for each data point.

---

[1]MNIST images are re-scaled to 32x32 images with a 0 padding.

Table 3: Neural networks used for the encoder and decoders of VAEs in the benchmarks

| | MNIST [CIFAR10] | SVHN | CELEBA | OASIS |
|---|---|---|---|---|
| ENCODER | (1[3], 32, 32) | (3, 32, 32) | (3, 64, 64) | (1, 208, 176) |
| LAYER 1 | CONV(128, (4, 4), STRIDE=2) BATCH NORMALIZATION RELU | LINEAR(1000) RELU | CONV(128, (5, 5), STRIDE=2) BATCH NORMALIZATION RELU | CONV(64, (5, 5), STRIDE=2) RELU |
| LAYER 2 | CONV(256, (4, 4), STRIDE=2) BATCH NORMALIZATION RELU | LINEAR(500) RELU | CONV(256, (5, 5), STRIDE=2) BATCH NORMALIZATION RELU | CONV(128, (5, 5), STRIDE=2) RELU |
| LAYER 3 | CONV(512, (4, 4), STRIDE=2) BATCH NORMALIZATION RELU | LINEAR(500, 16) | CONV(512, (5, 5), STRIDE=2) BATCH NORMALIZATION RELU | CONV(256, (5, 5), STRIDE=2) RELU |
| LAYER 4 | CONV(1024, (4, 4), STRIDE=2) BATCH NORMALIZATION RELU | - | CONV(1024, (5, 5), STRIDE=2) BATCH NORMALIZATION RELU | CONV(512, (5, 5), STRIDE=2) RELU |
| LAYER 5 | LINEAR(4096, 16) | - | LINEAR(16384, 64) | CONV(1024, (5, 5), STRIDE=2) RELU |
| LAYER 6 | - | - | - | LINEAR(4096, 16) |
| DECODER | (16 [32]) | (16) | (64) | (16) |
| LAYER 1 | LINEAR(65536) RESHAPE(1024, 8, 8) | LINEAR(500) RELU | LINEAR(65536) RESHAPE(1024, 8, 8) | LINEAR(65536) RESHAPE(1024, 8, 8) |
| LAYER 2 | CONVT(512, (4, 4), STRIDE=2) BATCH NORMALIZATION RELU | LINEAR (1000) RELU | CONVT(512, (5, 5), STRIDE=2) BATCH NORMALIZATION RELU | CONVT(512, (5, 5), STRIDE=(3, 2)) RELU |
| LAYER 3 | CONVT(256, (4, 4), STRIDE=2) BATCH NORMALIZATION RELU | LINEAR(3072) RESHAPE(3, 32, 32) SIGMOID | CONVT(256, (5, 5), STRIDE=2) BATCH NORMALIZATION RELU | CONVT(256, (5, 5), STRIDE=2) RELU |
| LAYER 4 | CONVT(3, (4, 4), STRIDE=1) BATCH NORMALIZATION SIGMOID | - | CONVT(128, (5, 5), STRIDE=2) BATCH NORMALIZATION RELU | CONVT(128, (5, 5), STRIDE=2) RELU |
| LAYER 5 | - | - | CONVT(3, (5, 5), STRIDE=1) BATCH NORMALIZATION SIGMOID | CONVT(64, (5, 5), STRIDE=2) RELU |
| LAYER 6 | - | - | - | CONVT(1, (5, 5), STRIDE=1) RELU |

Table 4: Neural Network used for the classifier in Sec. C.3

| OASIS CLASSIFIER | |
|---|---|
| INPUT SHAPE | (1, 208, 176) |
| LAYER 1 | CONV(8, (3, 3), STRIDE=1) BATCH NORMALIZATION LEAKYRELU MAXPOOL(2, STRIDE=2) |
| LAYER 2 | CONV(16, (3, 3), STRIDE=1) BATCH NORMALIZATION LEAKYRELU MAXPOOL(2, STRIDE=2) |
| LAYER 3 | CONV(32, (3, 3), STRIDE=2) BATCH NORMALIZATION LEAKYRELU MAXPOOL(2, STRIDE=2) |
| LAYER 4 | CONV(64, (3, 3), STRIDE=2) BATCH NORMALIZATION LEAKYRELU MAXPOOL(2, STRIDE=2) |
| LAYER 5 | LINEAR(256, 100) RELU |
| LAYER 6 | LINEAR(100, 2) SOFTMAX |

## E    Dataset size sensibility on SVHN

In Figure 7, we show the same plot for SVHN as in Sec. 5.2. Again the proposed method appears to be part of the most robust generation procedures to dataset size changes.

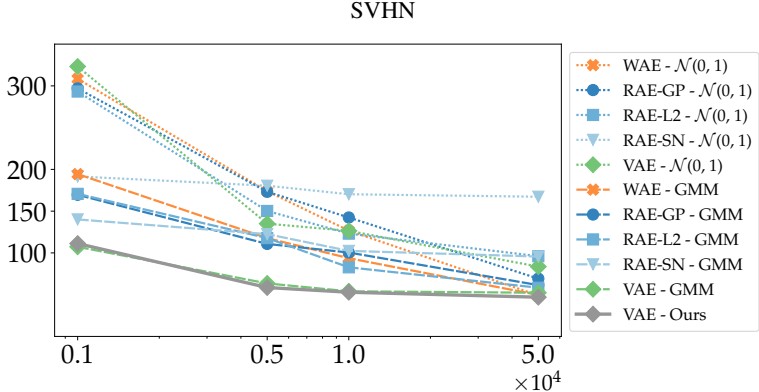

Figure 7: FID score evolution according to the number of training samples.

## F Ablation study

### F.1 Influence of the number of centroids in the metric

In order to assess the influence of the number of centroids and their choice in the metric in Eq. (**??**), we show in Figure 8 the evolution of the FID according to the number of centroids in the metric (left) and the variation of FID according to the choice in the centroids (right). As expected, choosing a small number of centroids will increase the value of the FID since it reduces the variability of the generated samples that will remain *close* to the centroids. Nonetheless, as soon as the number of centroids is higher than 1000 the FID score is either competitive or better than peers and continues decreasing as the number of centroids increases.

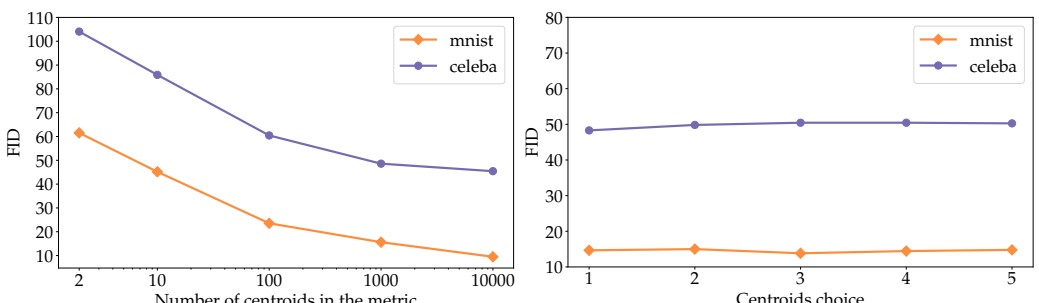

Figure 8: *Left:* FID score evolution according to the number of centroids in the metric (Eq. (**??**)). *Right*: The FID variation with respect to the choice in centroids. We generate 10000 samples by selecting each time different centroids ($k = 1000$).

To assess the variability of the generated samples, we propose to analyze some generated samples when only 2 centroids are considered. In Figure 9, we display on the left the decoded centroids along with the closest image to these decoded centroids in the train set. On the right are presented some generated samples. We place these samples in the top row if they are closer to the first decoded centroid and in the bottom row otherwise. Interestingly, even with a small number of centroids the proposed sampling scheme is able to access to a relatively good diversity of samples. These samples are not simply resampled train images or a simple interpolation between selected centroids as some of the generated samples have attributes such as glasses that are not present in the images of the decoded centroids.

Decoded centroid  Nearest train image                    Generated samples

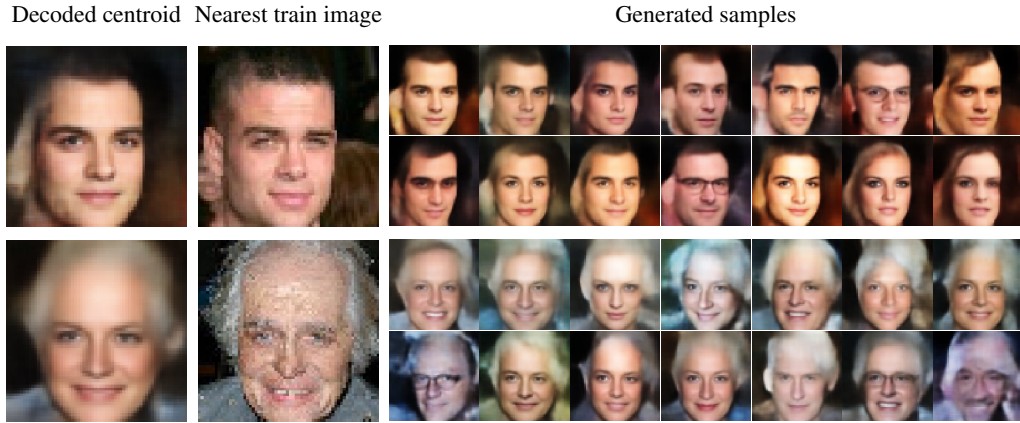

Figure 9: Variability of the generated samples when only two centroids are considered in the metric. *Left:* The image obtained by decoding the centroids. *Middle*: The nearest image in the train set to the decoded centroids. *Right:* Some generated samples. Each generated sample is assigned to the closest decoded centroid (top row for the first centroid and bottom row for the second one).

 **F.2 Influence of $\lambda$ in the metric**

In this section, we also assess the influence of the regularization factor $\lambda$ in Eq. (**??**) on the resulting sampling. To do so, we generate 10k samples using the proposed method on both MNIST and CELEBA datasets for values of $\lambda \in [1e^{-6}, 1e^{-4}, 1e^{-2}, 1e^{-1}, 1]$. Then, we compute the FID against the test set. Each time, we consider $k = 1000$ centroids in the metric. As shown in Figure 10, the influence of $\lambda$ remains limited. In the implementation, a typical choice for $\lambda$ is $1e^{-2}$.

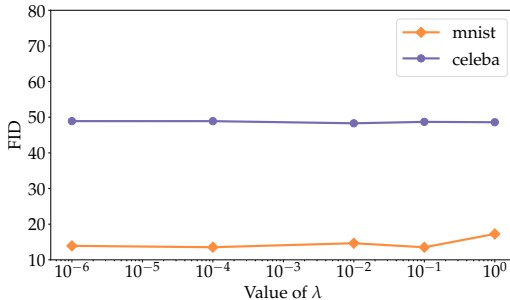

Figure 10: FID score evolution according to the value of $\lambda$ in the metric (Eq. (**??**)).

**F.3 The choice of $\rho$**

In the experiments presented, the smoothing factor $\rho$ in Eq. (**??**) is set to the value of the maximum distance between two closest centroids $\rho = \max_{i} \min_{j \neq i} \|c_j - c_i\|_2$. This choice is motivated by the fact that we wanted to build a smooth metric and so ensure some *smoothness* of the manifold while trying to interpolate faithfully between the metric tensors $\mathbf{G}_i = \mathbf{\Sigma}(x_i)^{-1}$. In particular, a too small value of $\rho$ would have allowed disconnected regions and the sampling may have not prospected well the learned manifold and would have only become a resampling of the centroids. On the other hand, setting a high value for $\rho$ would have biased the interpolation and the value of the metric at a $\mu(x_i)$. As a result, $\mathbf{G}(\mu(x_i))$ might have been very different from the one observed $\mathbf{\Sigma}(x_i)^{-1}$ since the other $\mu(x_j)$ would have had a strong influence on its value. The proposed value for $\rho$ appeared to work well in practice.

# G Can the method benefit more recent models ?

Our method proposes to build a Riemannian metric using the covariances in the posterior distributions. Thus, it can be easily plugged into more recent models provided that they have a Gaussian posterior distribution. In order to assess how it would benefit to more recent VAE models, we train a VAMP-VAE [19], a VAEGAN [11], an Adversarial AE [13] and an IWAE [1] and compare the generation FID obtained 1) with the prior or 2) when plugging our method. For this experiment, we conduct a hyper-parameter search consisting in training each model with 10 different configurations. For the VAMP we vary the number of pseudo-inputs in {10, 20, 30, 50, 100, 150, 200, 250, 300, 500}. For the VAEGAN, we use a discriminator similar to the encoder described in Table. 3 and vary the layer depth considered for the reconstruction loss in {2, 3, 4} and the factor balancing reconstruction/generation for the decoder's loss in {0.3, 0.5, 0.7, 0.8, 0.9, 0.99, 0.999}. For the AAE, we change the factor balancing the reconstruction loss and the regularization in {0.001, 0.01, 0.1, 0.25, 0.5, 0.75, 0.9, 0.95, 0.99, 0.999}. Finally, for the IWAE, we vary the number of importance samples in {2, 3, 4, 5, 6, 7, 8, 9, 10, 12}. For each model and generation scheme, we report the results of the model achieving the lowest FID on the validation set. According to Table. 5, the proposed generation method seems to benefit these models in almost all cases since the FID decreases when compared to the prior-based generation.

Table 5: FID (lower is better) vs. the test set using either the prior (classic approach) or by plugging our generation method.

| MODEL | GENERATION | MNIST | CELEBA |
|-------|-----------|-------|--------|
| VAMP | PRIOR | 34.5 | 67.2 |
| | OURS | **32.7** | **60.9** |
| IWAE | PRIOR | **32.4** | 67.6 |
| | OURS | 33.8 | **60.3** |
| AAE | PRIOR | 19.1 | 64.8 |
| | OURS | **11.7** | **51.4** |
| VAEGAN | PRIOR | 8.7 | 39.7 |
| | OURS | **6.1** | **31.4** |

Another approach that is interesting to compare to is the 2-stage VAE model proposed in [5]. Our method can indeed be seen as part of the methods trying to counterbalance the poor expressiveness of the prior distribution. In [5], the authors argue that the actual distribution of the latent codes (i.e. the aggregated posterior) is "likely not close to a standard Gaussian distribution" [5] leading to a distribution mismatch degrading the generation capability of the model. To address this issue, they propose to use a second VAE to estimate the learned distribution of the latent variables. Our approach starts with the same observation that the latent codes have no reason to follow the prior. However, it differs since we propose to adopt a fully geometric perspective and propose instead a sampling scheme using the intrinsic uniform distribution defined on the learned Riemannian manifold.

We nonetheless compare our method with models obtained with the official implementation provided by the authors of [5] on MNIST and CELEBA. To allow a fair comparison, we simply plug our method to the obtained trained models and build the metric using the posteriors coming from the $1^{st}$ stage VAE. In Table. 6, we compare the FID obtained 1) with the first stage VAE (*i.e.* prior), 2) with the second stage VAE [5] and 3) with our method. Again, our proposed generation method allows to achieve lower FID results.

Table 6: FID (lower is better) vs. the test set using the 2-stage VAE implementation [5] for either the reconstructed samples (recon.), using the prior ($1^{st}$ stage), using the 2-stage approach ($2^{nd}$ stage) or by plugging our generation method.

| DATASET | NETS | RECON. | $1^{st}$ STAGE | $2^{nd}$ STAGE | OURS |
|---------|------|--------|----------|----------|------|
| MNIST | SIMILAR TO [4] | 14.8 | 20.0 | 12.9 | **9.9** |
| CELEBA | SIMILAR TO [4] | 44.9 | 67.8 | 53.3 | **49.6** |
| CELEBA | SIMILAR TO [18] | 34.3 | 70.8 | 40.7 | **37.9** |