# OpenReview forum: "A Geometric Perspective on Variational Autoencoders"
_NeurIPS.cc/2022/Conference — NeurIPS 2022 Accept_

### Official Review · Reviewer_AaFY · 2022-07-03

**Rating:** 6
**Confidence:** 2
**Soundness:** 2 fair
**Presentation:** 2 fair
**Contribution:** 2 fair

**Summary:**

Authors identify a problem with the standard sampling process of a VAE. They then interpret the mean and variance predicted by a vanilla VAE encoder as the tangent space of a Riemannian Manifold. Under this interpretation, the authors then sample uniformly from the space.

Unfortunately, I do not have a good understanding of Riemannian manifolds, so I cannot evaluate the validity of the method described by the authors (if the U_Riem is indeed a valid distribution, for example).

However, assuming the validity of this method, it is important to measure empirically how good the samples derived from this method are under several metrics and datasets. The authors perform several experiments to this end in the Experiments section.


**Questions:**


Have experiments been performed with pre-trained models that already perform comparably with SOTA models, and does the new sampling method work better on top of those results?


**Limitations:**


A better sampling method for image generation would have wider implications on misinformation in the public sphere. However, I believe the empirical results are not strong enough to substantiate this as a worry for this particular work at the moment.

**Strengths And Weaknesses:**


Strengths:

I am not fully certain of this, but to my knowledge, this is a novel way of interpreting the latent space of the VAE, and leveraging the encoder predictions for improving sampling.

Weaknesses:

The authors would make a better case of their method if they took an _existing_ trained model that already does well, and show that their method of sampling achieves better samples.

Relatedly, the baselines appear to be implemented by the authors. While I understand this is for an “apples to apples” comparison against equivalent latent dimension sizes and architectures, it is hard to know for sure that the baselines will do better under some hyperparameter tuning, thereby weakening the strength of the results

Minor: the language could use a second pass, some grammatical errors:
- Line 17: “Nonetheless, when taken in their simplest version, it was noted that these models produce blurry samples most of the time.”
- Line 20: “Second, the prior distribution used…” run on sentence. -
- Line 34: “We show that this procedure significantly improves the generation..”
- Line 261: “This supports the fact that even the simplest VAE model”
- List is non-exhaustive

---

> ### Author Response · Authors · 2022-07-30
> **Thank you for your review!**
>
> We thank Reviewer AaFY for his/her review and suggestions to improve the experimental results.
>
> - *baselines implementations*: As stated in Appendix. D, in the paper, we decided to use the code and hyper-parameters provided by the authors (if available) for each method we compared to. If the code was not available we re-implemented the method.
>
> - *Experiments - peers hyper-parameter tuning*: We agree with the reviewer that an extensive hyper-parameter search would strengthen the results. Hence, following the reviewer’s advice, we perform a hyper-parameter search for the models considered in the paper i.e. WAE [1], VAMP [2], RAE-GP  [3] and RAE-L2 [3]. Since both HVAE and RHVAE models have a very time consuming training, we propose to replace those approaches with models having the same objective (i.e. enriching the posterior distribution). Do to so we consider a VAE with inverse autoregressive flows [4] (VAE-IAF) and a VAE with normalizing flows with radial/planar invertible transformations [5] (VAE-NF).
>
>     We train these models with 10 different hyper-parameter configurations on MNIST and CELEBA. We then show the generation vs. test FID of the model achieving the lowest FID on the validation set. Following the reviewer’s suggestion, we perform the experiment twice each time with a different neural network architecture (CNN or ResNet).
>
>     | Models | MNIST | CELEBA | MNIST | CELEBA |
>     |:---|:---:|:---:|:---:|:---:|
>     | Nets | CNN | CNN | Res | Res |
>     | AE - N(0,1) | 46.4 | 64.6 | 221.8 | 275.0 |
>     | WAE | 18.9 | 54.6 | 20.3 | 67.1 |
>     | VAE - N(0,1) | 40.7 | 64.1 | 47.8 | 69.5 |
>     | VAMP | 34.0 | 56.0 | 34.5 | 67.2 |
>     | VAE-NF | 29.3 | 55.4 | 32.5 | 67.1 |
>     | VAE-IAF | 27.5 | 56.5 | 30.6 | 66.2 |
>     | AE - GMM | 9.6 | 56.1 | 11.0 | 57.4 |
>     | RAE-GP | 9.4 | 52.5 | 11.4 | 59.0 |
>     | RAE-L2 | 9.1 | 54.5 | 11.5 | 58.3 |
>     | VAE - GMM | 13.1 | 55.5 | 12.4 | 59.9 |
>     | Ours | **8.5** | **48.7** | **10.7** | **53.2** |
>
> - *Experiments with pre-trained models*: In the paper, we retrained the models from scratch but we agree that the method can be applied to pre-trained models as well. For instance, we show in Reviewer 7Rkz response that the method can be easily plugged in more recent VAE models and allow these models achieving better generation results. For instance, it allows to decrease the test generation FID of a VAEGAN [6] from 39.7 to 31.4. As to current SOTA models, unfortunately, given the limited time frame, the compute they require and their complex architectures (hierarchical models) we did not have the time to try integrating this method in these models.
>
> Do not hesitate if you have any further questions.
>
>
> [1] Ilya Tolstikhin et al., Wasserstein autoencoders. ICLR, 2018
>
> [2] Jakub Tomczak and Max Welling. Vae with a vampprior. AISTATS, 2018.
>
> [3] Partha Ghosh, et al., “From variational to deterministic autoencoders”. ICLR 2020
>
> [4] Kingma, D. P., et al. “Improved variational inference with inverse autoregressive flow”. NIPS 2016
>
> [5] Rezende, D., & Mohamed, S. “Variational inference with normalizing flows”. ICML 2015.
>
> [6] Larsen, A. B. L., et al. “Autoencoding beyond pixels using a learned similarity metric”. ICML 2016

---

### Official Review · Reviewer_9492 · 2022-07-10

**Rating:** 6
**Confidence:** 2
**Soundness:** 3 good
**Presentation:** 3 good
**Contribution:** 2 fair

**Summary:**

The paper takes a geometric view on the classic VAE model by reinterpreting the latent space through the lens of Riemannian manifolds. It proposes a new sampling technique utilizing the learned manifold and Riemannian metric and empirically shows the method quantitatively improves sample quality on several datasets over various standard VAE baselines.

**Questions:**

**Suggestions**:
- 1, line 22-23: perhaps change ‘inducing over-regularization’ with ‘manifold/distribution mismatch’
- 4.3, line 182-185: it is stated that the Beta term to weigh the influence of the KL divergence on the total loss is predominantly seen as a reguralizer, similar to [18]. And that ‘the proposed vision [is kept] and [not amended] for training’. It would increase clarity to briefly touch upon the fact that while the ultimate outcome of the KL-loss as a regularizar is similar, the justification of [18] is very much different.
- 5.1: given the existence of appendix D, perhaps transfer the description of baseline models and benchmark datasets here to focus the section on results discussion. Splitting up the discussion between paragraphs separated by a line of white space would likely make it easier for the reader to grasp the main experimental take-aways.

**Possible Typos / formatting suggestions**:
- Abstract, line 2: *we indeed argue..* → we argue
- 1, line 13: *map into* → map to
- 1, line 17: *these models produce .. samples* → these models produce blurry samples on image generation tasks most of the time
- 1, line 35: *improves significantly…* → improves the generation process from a vanilla VAE significantly
- 1, line 36: *the training ..* → the training. The proposed sampling method outperforms more advanced VAE …
- 4.3, line 133: *voluntarily chosen* → (often) chosen?
- 4.3, line 134: *Hence, the … as follows.* → Hence, the .. as follow:
- 4.3, line 144: *As first approximation and since … the VAE model* → Since …, as a first approximation the VAE model
- 4.3, line 147: *This simplified drastically* → This drastically simplified
- 4.3, line 182: *that, likewise [18],* → that like [18]
- 4.6, line 202: *may be easily appended in Figure* → can be observed/seen in Figure
- 4.6, line 231: *below are presented….* → refer to figure number
- 6, line 309: *Then, we proposed* → We proposed
- 6, line 311: *It showed to be competitive* → The proposed method was empirically shown to be competitive


**Limitations:**

The authors are careful to highlight certain practical ‘shortcuts’ of their method where it deviates from theory and the consequences, e.g. section 4.3 starting line 170. They also spent considerable effort in the various appendices, e.g. appendix B and F, to further provide practical implementation details/considerations of their method. While the potential negative societal impact of this specific work seems small, it is not entirely clear why the authors highlight appendix C as answering this question. The only line that seems relevant here is C.3. 100-102?

**Strengths And Weaknesses:**

**Strengths:**
- [**quality**] the elaborate background on Riemannian geometry and extensive appendices nicely introduce and illustrate the proposed solution. The technical derivations are also clearly presented and well written. The empirical results show improvements over classic VAE baselines.
- [**originality**] while there are increasingly many works around enriching the VAE framework by extending it to use more complex distributions, e.g. normalizing flows, or adding distributions defined on manifolds with a non-trivial geometry, e.g. hyperspherical/hyperbolic, the angle proposed in this work adds a Riemannian geometry perspective in a novel and empirically useful manner.

**Weaknesses**:
- [**clarity**] the at times non-standard English makes this paper sometimes challenging to read. Other formatting choices could likely strengthen readability as well, e.g. section 5. (see next section for suggestions on possible sentences, subsections that could perhaps be improved upon).
- [**significance**] as noted by the authors in the introduction, one of the major reasons for the classic VAE’s sustained appeal for modern machine learning tasks is its simplicity and ease of use. This surely contributes to it often still having an edge over more ‘recent methods’, e.g. diffusion based models, with higher quality results. As such, the complexity of the proposed method is likely to limit its adaptation, especially if it can’t be shown to perform on par with state of the art methods.

---

> ### Author Response · Authors · 2022-07-30
> **Thank you for your review!**
>
> We thank Reviewer 9492 for his/her detailed review of our paper and suggestions for improving its clarity. Please see below our answers to the points you raised.
>
> - *Significance*: We agree with the reviewer that in terms of image generation, many recent works focusing on diffusion models have shown exciting results. However, we show in **Reviewer 7Rkz** response that the method can be easily plugged in more recent VAE models and benefit these models in terms of image generation results leading to potential improvements of VAE-based methods. For instance, it allows to decrease the test generation FID of a VAEGAN [1] from 39.7 to 31.4. Hence, we think that a better understanding of the latent space of VAE models can lead to higher quality generations.  Moreover, VAE models through their latent space can also allow very interesting downstream tasks such as clustering, classification, data reconstruction or allow to perform meaningful interpolations. Nonetheless, this latent structure remains poorly understood and we hope that this new geometry-based vision of the model will lead to new original methods to enhance those tasks as well.
> - *Non standard English*: Thank you very much for spotting the non-standard English expressions and proposing these changes. We have amended the paper accordingly.
> - *l.182/185*: We have slightly changed l.182/185 to better reflect the point of view of [2] as to the role of the KL in the VAE loss.
> - *Section 5.1 clarity*: We agree that Section 5.1 can be divided into several paragraphs to improve readability. We have changed this in the revised version.
> - *Ethical concerns*: We indeed highlight Appendix C when answering this question of the checklist since to us the method itself does not major ethical concerns but using it for on medical image as we did in Appendix C may require cautious.
>
> Do not hesitate if you have any further questions.
>
> [1] Larsen, A. B. L., et al. “Autoencoding beyond pixels using a learned similarity metric”. ICML 2016
>
> [2] Partha Ghosh, et al., “From variational to deterministic autoencoders”. ICLR 2020

---

### Official Review · Reviewer_7Rkz · 2022-07-11

**Rating:** 7
**Confidence:** 3
**Soundness:** 3 good
**Presentation:** 3 good
**Contribution:** 3 good

**Summary:**

This paper proposes to use Riemannian geometry to analyze the latent space of variational autoencoders (VAE).  Specifically, the authors define a Riemannian Gaussian distribution on the learned latent space. Based on this, a novel sampling process is proposed in which samples from the uniform distribution are defined on the learned Riemannian latent space. Experiments show that the proposed sampling method achieves competitive results on several benchmark datasets.

**Questions:**

1. It would be interesting to see some discussion about how the proposed method can benefit other more recent VAE-based methods rather than only the vanilla VAE.

**Limitations:**

The authors adequately addressed the limitations and potential negative societal impact of their work

**Strengths And Weaknesses:**

Strengths:

1. This paper is well written and easy to follow. The authors clearly explained the mathematics behind the geometrical interpretation of the VAE framework. It is very helpful that the authors give a brief review of Riemannian geometry and the Riemannian Gaussian distribution.

2. Several experiments with different metrics show that the proposed method is effective and robust.


Weaknesses:

1. While I understand that this work focuses on the sampling process for vanilla VAE without modifying the training procedure, I think this sampling process can be seen as a second stage for estimating the learned latent distribution of vanilla VAE. For comparison, in two-stage VAE [1],  they achieve 34 for FID on CelebA with the second stage of training which is approximating the learned latent distribution by vanilla VAE.  Some discussion/comparison between these methods would be helpful.

[1] Dai, Bin, David Wipf, and Minhao Jiang. "Diagnosing and Enhancing VAE models (ICLR 2019)."

---

> ### Author Response · Authors · 2022-07-30
> **Thank you for your review!**
>
> We thank Reviewer 7Rkz for his/her careful review of our paper and suggestions to improve it. Please see below the answers to your questions.
>
> - *Comparison with a 2-stage VAE*: We agree with the reviewer that our method can be seen as part of the methods trying to counterbalance the poor expressiveness of the prior distribution. In [1], the authors argue that the actual distribution of the latent codes  (i.e. the aggregated posterior) is "likely not close to a standard Gaussian distribution" [1]. Hence, they propose to use a second VAE to estimate the learned distribution of the latent variables. Our approach starts with the same observation that the latent codes have no reason to follow the prior. However, it differs since we propose to adopt a fully geometric perspective and propose instead a sampling scheme using the intrinsic uniform distribution defined on the learned Riemannian manifold.
>
>
>     As suggested by the reviewer, we nonetheless compare our method with models obtained with the official implementation provided by the authors of [1] on MNIST and CELEBA. To allow a fair comparison, we simply plug our method to the obtained trained models and build the metric using the posteriors coming from the 1st stage VAE. We compare the FID obtained 1) with the first stage VAE, 2) with the second stage VAE and 3) with our method. We were not able to reproduce a FID of 34 on CELEBA with the provided implementation but we were still able to reduce the FID to 40.7 for the second stage sampling and leading to 37.9 for our method.
>
>     | Dataset | Nets | Latent dim | Recon | 1st stage | 2nd stage | Ours |
>     |:---:|:---:|:---:|:---:|:---:|:---:|:---:|
>     | MNIST | Similar to [2] | 16 | 14.8 | 20.0 | 12.9 | **9.9** |
>     | CELEBA | Similar to [2] | 64 | 44.9 | 67.8 | 53.3 | **49.6** |
>     | CELEBA | Similar to [3] | 64 | 34.3 | 70.8 | 40.7 | **37.9** |
>
>
> - *Can it benefit to more recent models*: Our method proposes to build a Riemannian metric using the covariances in the posterior distributions. Thus, it can be easily plugged into more recent models provided that they have a Gaussian posterior distribution. Hence, we trained a VAMP-VAE [4], a VAEGAN [5], an Adversarial AE [6] and an IWAE [7] and compared the generation FID obtained 1) with the prior or 2) when plugging our method. Following Reviewer AaFY, we conducted a hyper-parameter search for these models consisting in training each model with 10 different configurations. For each model and generation scheme, we report the results of the model achieving the lowest FID on the validation set. According to the table, the proposed generation method seems also to benefit these models in almost all cases since the FID decreases when compared to the prior-based generation.
>
>     | Models | MNIST | CELEBA |
>     |:---:|:---:|:---:|
>     | VAMP – prior | 34.5 | 67.2 |
>     | VAMP – ours | **32.7** | **60.9** |
>     | IWAE – prior | **32.4** | 67.6 |
>     | IWAE – ours | 33.8 | **60.3** |
>     | AAE – prior | 19.1 | 64.8 |
>     | AAE – ours | **11.7** | **51.4** |
>     | VAEGAN - prior | 8.7 | 39.7 |
>     | VAEGAN – ours | **6.1** | **31.4** |
>
> Do not hesitate if you have any further questions.
>
> [1] Dai, Bin, David Wipf, and Minhao Jiang. "Diagnosing and Enhancing VAE models”. ICLR, 2019
>
> [2]  InfoGAN: Interpretable representation learning by information maximizing generative adversarial nets. NIPS, 2016.
>
> [3] Ilya Tolstikhin et al., Wasserstein autoencoders. ICLR, 2018
>
> [4] Jakub Tomczak and Max Welling. Vae with a vampprior. AISTATS, 2018.
>
> [5] Larsen, A. B. L., et al. “Autoencoding beyond pixels using a learned similarity metric”. ICML 2016
>
> [6] Burda, Y., et al. “Importance Weighted Autoencoders”. ICLR 2016.

---

### Author Response · Authors · 2022-07-30
**Thank you for your reviews!**

First of all, we would like to sincerely thank all the reviewers for taking the time to carefully read and review our paper and for their constructive and positive comments. We will respond to each reviewer in separate comments but here is a summary of the changes we made in the revised version and the responses we made to the reviewers.

- **Paper changes**:
    - Further to Reviewer 7Rkz comments we have added Appendix G where we provide some discussion and empirically show how the proposed generation process benefits more recent VAE-based methods. We refer to this Appendix in the introduction and Sec. 4.5. In this section we also discuss and compare our method with the 2-stage VAE approach proposed in [1]
    - We correct the typos and grammatically incorrect sentences highlighted by Reviewer 9492 and Reviewer AaFY.
    - Following Reviewer 9492’s suggestions we change Sec.5.1 and move the model description to Appendix D and split the section into paragraphs to improve readability.
    - Further to Reviewer AaFY’s suggestions, we add Appendix C.4 where we additionally perform a hyper-parameter search for the models considered in the paper  a hyper-parameter search on MNIST and CELEBA for the models considered in the paper to strengthen the comparison


- **Reviewer 7Rkz responses summary**:
    - We provide some discussion between the 2-stage VAE model proposed in [1] and our method. We also compare these approaches using the implementations provided by the authors on MNIST and CELEBA.
    - We explain how the method can be integrated into more recent VAE-based models and show empirically that it can benefit these models by increasing the generation performance in terms of FID. For example, applying the method to a VAEGAN [2] allows achieving a FID of 31.4 vs. 39.7 with the prior.

- **Reviewer 9492 responses summary**:
    - We propose to correct the typos and grammatically incorrect sentences highlighted by the reviewer.
    - Further to the reviewer suggestion, we agree to reformulate l.182-185 and split Sec.5.1 into several paragraphs for clarity.
    - As for Reviewer 7Rkz, we discuss and show how the method benefits more recent VAE models.

- **Reviewer AaFY responses summary**:
    - We clarify that we used the code and hyper-parameters (if available) provided by the authors in the experiments of the paper.
    - Further to the reviewer's suggestion, we additionally perform a hyper-parameter search for the models considered in the paper consisting in training the models with 10 different configurations and 2 different neural networks to strengthen the comparison. Due to the high training time of both the HVAE and RHVAE models, we propose to replace those approaches with models having the same objective (i.e. enriching the posterior distribution) such as a VAE with inverse autoregressive flows [3] (VAE-IAF) and a VAE with normalizing flows with radial/planar invertible transformations [4] (VAE-NF). We show the generation vs. test FID on MNIST and CELEBA and the proposed method still achieves the best results.

In any case, do not hesitate if you have any questions.

Best,

The authors,

[1] Dai, Bin, David Wipf, and Minhao Jiang. "Diagnosing and Enhancing VAE models (ICLR 2019)."

[2] Larsen, A. B. L., et al. “Autoencoding beyond pixels using a learned similarity metric”. ICML 2016

[3] Kingma, D. P., et al. “Improved variational inference with inverse autoregressive flow”. NIPS 2016

[4] Rezende, D., & Mohamed, S. “Variational inference with normalizing flows”. ICML 2015.

---

### Meta-Review · Area_Chair_4JqU · 2022-08-26

**Recommendation:** Accept
**Confidence:** Certain

**Metareview:**

The submission proposes a geometrically motivated method to sample from a trained variational autoencoder by defining a Riemannian metric on the latent space and a corresponding distribution to correct the sampling process of VAEs. According to most reviewers and reading the submission, the method is clearly explained and provides a consistent improvement in sample quality according to FID. The authors have mostly addressed the reviewers concerns as well.
I recommend this paper for acceptance.

**Award:**

No

---

### Decision · Program_Chairs · 2022-09-14

Accept